



# Thermocline depth and euphotic zone thickness regulate the abundance of diazotrophic cyanobacteria in Lake Tanganyika

Benedikt Ehrenfels[1,2], Maciej Bartosiewicz[3], Athanasio S. Mbonde[4], Kathrin B.L. Baumann[1,2], Christian Dinkel[1], Julian Junker[5,6], Tumaini Kamulali[4], Ismael A. Kimirei[4,7], Daniel Odermatt[1], Francesco Pomati[8], Emmanuel A. Sweke[4,9], Bernhard Wehrli[1,2]

[1]Eawag, Swiss Federal Institute of Aquatic Science and Technology, Department Surface Waters – Research and Management, Kastanienbaum, Switzerland
[2]ETH Zurich, Institute of Biogeochemistry and Pollutant Dynamics, Zurich, Switzerland
[3]University of Basel, Department of Environmental Sciences, Basel, Switzerland
[4]TAFIRI, Tanzania Fisheries Research Institute, Kigoma, Tanzania
[5]Eawag, Swiss Federal Institute of Aquatic Science and Technology, Department Fish Ecology and Evolution, Kastanienbaum, Switzerland
[6]University of Bern, Institute of Ecology & Evolution, Bern, Switzerland
[7]TAFIRI, Tanzania Fisheries Research Institute, Kigoma, Tanzania
[8]Eawag, Swiss Federal Institute of Aquatic Science and Technology, Department Aquatic Ecology, Dübendorf, Switzerland
[9]DSFA, Deep Sea Fishing Authority, Zanzibar, Tanzania

*Correspondence to*: Benedikt Ehrenfels (benedikt.ehrenfels@eawag.ch)

**Abstract.** In spite of the fact that cyanobacterial blooms are classically associated with high nutrient loadings, there is also abundant evidence revealing that nitrogen fixing cyanobacteria (diazotrophs) can prevail under oligotrophic conditions. The mechanisms favouring diazotrophs in oligotrophic water bodies, however, remain poorly resolved. Here we analyse biogeochemical and ecological factors regulating the distribution of nitrogen fixing cyanobacteria in the oligotrophic Lake Tanganyika using sensor profiles of hydrodynamic conditions, nutrient and pigment analyses, as well as phytoplankton community assessment. During periods of stable or re-establishing water column stratification, we find evidence that the location of the thermocline and the euphotic depth can create a functional niche for diazotrophic cyanobacteria: Nitrogen limitation provides an ecological advantage for an apparent mutualistic interaction between diazotrophs and diatoms when the upward transport of nitrate into the euphotic zone is reduced by a subjacent thermocline. Diazotrophs, comprising the filamentous genera *Dolichospermum* and *Anabaenopsis*, are key players under these conditions (up to 41.7 % of phytoplankton community), while they are rare otherwise. By contrast, a thermocline located within the euphotic zone allows for the rapid vertical transport of nitrate for a thriving nitrate assimilating phytoplankton community that evidently outcompetes diazotrophs. Finally, multiple observations of relatively high diazotroph densities in the upwelling region in the South of Lake Tanganyika imply that they may additionally thrive under high nutrient conditions, when nitrogen is heavily deficient with respect to phosphorous. This study highlights that, under nitrogen deficient conditions, cyanobacterial blooms may form in response to reduced nutrient fluxes to the productive surface waters.



## 1 Introduction

Cyanobacterial blooms are a widespread phenomenon and increase in frequency as well as severity due to human-induced alterations of natural systems, such as eutrophication or climate change (Huisman et al., 2018; Paerl and Huisman, 2009). Classically, cyanobacterial blooms have been studied in the context of eutrophication of lakes and coastal seas. There is an ongoing debate regarding the exact factors triggering the formation of cyanobacterial blooms (Bartosiewicz et al., 2019; Downing et al., 2001; Glibert et al., 2018), but there is broad consensus that eutrophication is a prerequisite for their formation

in intensely human-influenced water bodies (Molot et al., 2014). While eutrophication studies have deepened our understanding of cyanobacterial bloom formation in relatively small oligotrophic lakes experiencing significant anthropogenic nutrient inputs (Almanza et al., 2019; Carey et al., 2008; Posch et al., 2012; Winter et al., 2011), the mechanisms allowing for cyanobacterial dominance in large, warm oligotrophic systems, such as the Northern Pacific Subtropical Gyre (NPSG), Gulf of Aqaba, or Lake Tanganyika, remain elusive.


The cyanobacterial blooms in those water bodies were, however, often regarded in expectation of a similar positive relationship between nutrient availability and cyanobacterial abundance (e.g. White et al., 2007). While nutrient injections into oligotrophic surface waters may induce phytoplankton blooms in general (e.g. McGillicuddy and Robinson, 1997), this mechanism might not necessarily apply to all blooms of cyanobacteria in particular. Indeed, contrary to those expectations, cyanobacterial blooms

in the NPSG are associated with highly stratified waters (Dore et al., 2008) and a deep thermocline in combination with declining solar irradiance (White et al., 2007) indicative of low nutrient fluxes to the sun-lit surface waters. Similarly, studies in the Gulf of Aqaba and Lake Tanganyika show that cyanobacteria can dominate the phytoplankton community during periods of pronounced stratification when nutrients are scarce (Cocquyt and Vyverman, 2005; Post, 2005). Despite such evidence, the paradigm that stratification in oligotrophic systems prevents cyanobacterial blooms still stands (e.g. Huisman et al., 2018).


These findings suggest, that some cyanobacterial blooms might in fact form in response to nutrient scarcity. The nitrogen (N) acquisition pathway of the bloom forming taxa is key to understanding cyanobacterial dominance under such conditions. In contrast to eutrophic waters, where bloom-forming cyanobacteria, such as *Microcystis* or *Planktothrix*, are capable of efficiently exploiting the excess nutrients (Huisman et al., 2018), cyanobacterial blooms in large, warm oligotrophic systems

are often formed by filamentous diazotrophs in association with diatoms (Foster and O'Mullan, 2008; Foster and Zehr, 2019; Van Meel, 1954). Diazotrophs fix atmospheric N, which allows them to thrive under the absence of a dissolved inorganic nitrogen (DIN) source. To grow in abundance, diazotrophs might require essential micronutrients or readily available carbon (Harke et al., 2019) from their partner, while the diatom symbiont profits from the freshly fixed N (Foster et al., 2011). Recent modelling work found that diazotroph-diatom associations is fittest under high irradiance and DIN deficiency, as long as the

concentrations of phosphate ($PO_4^{3-}$) and iron (Fe) are still sufficient (Follett et al., 2018). The supply of DIN to the sun-lit



surface zone, sustained by the upward transport from deep waters, may therefore exert significant bottom-up control over the abundance of diazotrophic cyanobacteria in oligotrophic waters, if no other limitation is apparent.

Based on these studies, we hypothesize that under reduced DIN fluxes into the euphotic zone, diazotrophs have an ecological advantage through the ability to access to both N and light, whereas DIN-assimilating taxa lack N at the surface and suffer from light limitation in the DIN-rich deep water. We tested this hypothesis in Lake Tanganyika, which provides a well-confined pelagic ecosystem with a severe N deficit. This large tropical lake is permanently stratified and characterized by a nutrient-rich hypolimnion overlaid by an oligotrophic epilimnion. Nutrient regeneration is maintained by seasonal mixing during the dry season (Plisnier et al., 1999; Verburg et al., 2011) with strongest upwelling intensities in the South and local upwelling events caused by internal seiches (Naithani et al., 2003) and Kelvin waves throughout the year (Naithani and Deleersnijder, 2004). Importantly, external inputs do not play a significant role in the nutrient budget of the lake (Langenberg et al., 2003) and N deficiency with respect to phosphorous (Edmond et al., 1993) provides a high potential for N fixing cyanobacteria. Blooms of the filamentous, diazotrophic cyanobacterium *Dolichospermum sp.* (formerly *Anabaena sp.*) have been reported to occur when the lake re-stratifies at the end of the dry season (Cocquyt and Vyverman, 2005; Hecky and Kling, 1981; Langenberg et al., 2002; Narita et al., 1986) as well as during the calm rainy season, when the lake is well-stratified (Descy et al., 2005; Salonen et al., 1999; Vuorio et al., 2003). To identify the physicochemical drivers and specific niches of these recurring blooms, we collected a combination of physical, biogeochemical, pigment, and phytoplankton diversity data during two lake-wide sampling campaigns at the end of dry season and the end of the rainy season and synthesized our key findings in two conceptual scenarios that might be applicable to other N deficient oligotrophic systems.

## 2 Material & Methods

### 2.1 Study site

The hydrodynamics of Lake Tanganyika, by volume the largest lake in the East African Rift System (Fig. 1), are controlled by seasonal and regional differences in surface meteorology. Verburg et al. (2011) distinguished four stages in the annual cycle: (1) The lake is stratified with a stagnant water column during the calm and warm rainy season from November to April. (2) At the beginning of May, cool, southerly trade winds push surface waters to the North tilting the thermocline and leading to upwelling in the South. The stratification is further weakened by heat loss to the atmosphere during the dry season from May to September. (3) The latitudinal gradient in heat exchange (Verburg et al., 2011) or the equilibrium flow of the surface water as a result of decreasing trade winds (Delandmeter et al., 2018), reverse the circulation pattern during the late dry season from July to September. (4) Once the trade winds cease in October, the lake circulation slows down and the thermocline gradually re-establishes and initiates oscillations through internal waves (Naithani et al., 2003) and secondary upwelling in the North (Plisnier et al., 1999). Variability in the circulation and stratification patterns as well as the magnitude of riverine inflow cause changes in the concentration of planktonic detritus as well as particular and dissolved organic matter in the upper water





column. These effects control the light attenuation and thus regulate the depth of euphotic zone (Plisnier et al., 1999) that was shown to vary between 20 and 70m, on a weekly to monthly timescale (Langenberg et al., 2002) with yearly averages of about 30-40 m (Cocquyt and Vyverman, 2005; Descy et al., 2005; Hecky et al., 1978).

## 2.2 Sampling

The dynamic physical variability in Lake Tanganyika and the complex logistics pose challenges for frequent observations. In order to combine good spatial coverage with a seasonal perspective, we collected vertical profiles along a lake-wide transect (9 stations) during two cruises at the end of the dry season (28 September - 8 October 2017) and the end of the rainy season (27 April - 7 May 2018) (Table S1). This sampling approach has been adopted in previous studies (e.g. Salonen et al., 1999; Stenuite et al., 2009). It allowed us to capture the pronounced North-South differences as well as to observe the *Dolichospermum* blooms typical for the end of the dry season and rainy season. During these two expeditions, hereafter 'Sep/Oct' and 'Apr/May', the CTD profiling (Sea-Bird SBE 19plus) and discrete water sampling at 5 to 25 m depth intervals (Niskin bottles, 20-30 L) was carried out on-board of M/V Maman Benita.

## 2.3 Physical parameters

The euphotic depth was defined as the depth, where 1 % of the surface photosynthetic active radiation arrives and is an often-used proxy for light limitation of photosynthesis in phytoplankton cells. Water column stability was calculated as buoyancy frequency ($N^2$) from CTD downcasts using the software SBE Data Processing. Clear peaks in buoyancy frequency were interpreted as thermoclines.

Only thermoclines limiting the DIN flux were relevant for our study, i.e. these separating DIN-depleted surface waters from underlying DIN-rich water masses. We refer to those as primary thermoclines in the following, while other thermoclines will be called secondary thermoclines. To compare the overall thermal stability of the upper water column among the two sampling occasions, we calculated the Schmidt stability over 1 $m^2$ in the upper 100 m for each station using the R package 'rLakeAnalyzer' (Winslow et al., 2019). The Schmidt stability (Idso, 1973; Schmidt, 1928) denotes the energy required to homogenize the water column. The temperature profiles pertaining to Apr/May are also presented by C. Callbeck, B. Ehrenfels, K.B.L. Baumann, B. Wehrli, and C.J. Schubert (manuscript in review at *Nat. Comms.*).

## 2.4 Nutrient concentrations

Nutrient subsamples were taken directly from the Niskin bottles and filtered sterile using 0.2 μm cellulose acetate membrane filters. The filtered samples (~50 mL) were stored cool before processing. Subsamples of 5 mL were pipetted into reaction tubes and reagents were added for each nutrient (phosphate: $PO_4^{3-}$; ammonium: $NH_4^+$; nitrate: $NO_3^-$; nitrite: $NO_2^-$) following standard methods (Grasshoff et al., 1999; Holmes et al., 2011; Schnetger and Lehners, 2014). Reagents and standards were prepared freshly each day. Ammonium was analysed with a Turner Trilogy fluorometer, while all other nutrients were



measured with a NOVA 60 photometer. The detection limits of these methods were 0.2, 0.4, 0.4, and 0.05 µmol/L on average for $PO_4^{3-}$, $NH_4^+$, $NO_3^-$, $NO_2^-$, respectively. We determined the detection limits using the regression coefficient and the residual standard deviation of the calibration line and excluded values from further analysis, if they were not markedly

different from the blank. We calculated the N deficit according to the Redfield stoichiometry of phytoplankton as $16 \cdot [PO_4^{3-}]$ - $[NH_4^+]$ - $[NO_3^-]$ - $[NO_2^-]$.

### 2.5 Pigments

Photopigment concentrations were analysed as quick proxies for photosynthetic activity of phytoplankton (chlorophyll *a*) as
well as cyanobacterial abundance (phycoerythrin and phycocyanin; Salonen et al., 1999). Samples for pigment analyses were filled in 20 or 30 L containers and stored in the dark. Chlorophyll *a* (Chl) concentrations were determined according to the recommendations of Wasmund, Topp and Schories (2006). Between 2 and 4 L were filtered onto 47 mm glass fibre filters (GF55, Hahnemühle) using a suction pump. The filters were immediately transferred to 15 mL plastic tubes and 5 mL ethanol (> 90 %) were added, followed by 10 min cold ultrasonification. The samples were stored cool overnight and sterile-filtered
using 0.2 µm cellulose acetate membrane filters on the next morning. The extracts were measured with a Turner Trilogy fluorometer. We followed the same procedure for measuring the concentrations of phycoerythrin (PE) and phycocyanin (PC) with the distinction of using phosphate buffer (0.05 M, pH neutral) as solvent. Samples were handled and processes in the dark. We calculated the integrals from the water surface to 125 m depth, to include only photosynthetically active organisms from the oxygenated epi- and metalimnion.


### 2.6 Phytoplankton community analyses

Phytoplankton samples were collected to validate the pigment proxies. Sampling focussed on the medium to large size fractions (> 10 µm) of the phytoplankton community to capture the bloom forming, diazotrophic cyanobacterium *Dolichospermum* and other associated taxa (Hecky and Kling, 1981) in sufficient densities. Briefly, phytoplankton samples
(4-10 L) were concentrated to 20 ml on a 10 µm plankton net and then immediately fixed with alkaline Lugol solution. 2 mL subsamples were then analysed by inverted microscopy (× 400 magnification). Phytoplankton genera and species were determined according to Komárek and Anagnostidis (1999, 2007) and Streble and Krauter (1988) and detailed taxonomic data are given in Table S2.

### 2.7 Breakpoint analysis

We performed a breakpoint analysis using the R package 'segmented' (Muggeo, 2008) to identify the location of a potential threshold values in the relationship between depth-integrated diazotroph abundance and the depth of the thermocline. For breakpoint estimation, the method uses a simple linearization technique and required exclusion of a single apparent outlier (station 9 in Sep/Oct). We additionally examined whether the relative position of the thermocline to the euphotic depth had an
effect on diazotroph abundance in Apr/May, whereas we had no irradiance data for such an analysis in Sep/Oct.



## 3 Results

### 3.1 Biogeochemical characterization

Our sampling in Sep/Oct was accompanied by southerly trade winds. By contrast, the weather was calm and rainy in April, whereas the onset of cool trade winds initiated the upward tilting of the thermocline in the southern basin towards the end of

the sampling in May (Fig. 1 & S1). Accordingly, Lake Tanganyika was more heavily stratified during Apr/May, as indicated by the steeper thermoclines (Fig. 1, S1 & Table S3), reaching $N^2$ values of up to $5.0 \cdot 10^{-4}$ $s^{-2}$ compared to a maximum of $2.7 \cdot 10^{-4}$ $s^{-2}$ in Sep/Oct. This is further supported by the higher overall water column stability, ranging from 5.8-8.3 kJ $m^{-2}$ in Apr/May as compared to 4.2-6.0 kJ $m^{-2}$ in Sep/Oct. Beside the primary thermocline, deeper, secondary thermoclines had formed just above the nitrate ($NO_3^-$) peak in Sep/Oct (Fig. 1 & Table S3). In Apr/May, this feature was observed only for

station 1, while near-surface secondary thermocline was recorded around 30 m depth at stations 3 and 4 (see supporting information sections 2 & 5).

The N deficit (98 % of all observations) persisting throughout the water column (Fig. S2) implies that primary productivity was N limited. Nitrate was the main form of DIN accessible to phytoplankton (Fig. 1), while $NH_4^+$ and $NO_2^-$ remained below

detection limit in the upper 120-150 m. The euphotic zone, characterized in Apr/May, varied between 35.8-54.8 m with an average of 47.3 m. Nitrate concentrations in the euphotic zone (Fig. 1 b,e) were also often below detection limit while $PO_4^{3-}$ concentrations were relatively high (Fig. 3 & S2).

The magnitude of the density gradient within the primary thermocline as well as its depth regulated the upward flux of

metalimnetic $NO_3^-$ (maximum: 65-135 m) into the overlaying euphotic zone (0-56 m). The primary thermocline was associated with the nutricline (strong vertical gradients in nutrient concentrations), separating $NO_3^-$-depleted surface waters from underlying $NO_3^-$-rich water masses. Hence, the vertical transport of $NO_3^-$ into the productive layer was efficient only if the primary thermocline was located within the euphotic zone. In Apr/May, this condition was fulfilled only at stations 1, 8, and 9. At stations 2-5, the primary thermocline was located well below the euphotic zone and at stations 6 and 7 near its lower

bound (Fig. 1 & 4b). We have no irradiance data from Sep/Oct, but the deepest positions of the primary thermocline (below 40 m) were found in the centre of the lake at stations 4-6.

### 3.2 Photopigment distribution

The Chl maximum was usually located between 30 and 50 m, i.e. at the bottom of the euphotic zone (Fig. 1 d,g & 3b).

Exceptions from this general pattern occurred in Apr/May when the primary thermocline was positioned below the euphotic zone (stations 2-6) (Fig. 1, 3a & S4) and to a lesser extend in Sep/Oct where the primary thermocline was below 40 m (stations 4-6). For these observations, the main or secondary Chl peak was located at or near the water surface and the PE and PC concentrations were also highest in surface waters (Fig. 1 b,f & S3). At other stations, the PE and PC concentrations were





generally low (< 0.1 µg PC L$^{-1}$ and < 0.001 µg PE L$^{-1}$, respectively). The concentrations of PE and PC were significantly

correlated with diazotroph abundance throughout the entire study (p < 0.001, Pearson correlation coefficients: 0.52-0.88). The depth-integrated concentrations of PE and PC were higher in Apr/May than in Sep/Oct (on average 17.37 versus 2.75 mg PC m$^{-2}$ and 0.42 versus 0.18 mg PE m$^{-2}$; Fig. S3).

### 3.3 Phytoplankton community

The phytoplankton community in Lake Tanganyika was dominated by chlorophytes, diatoms, and cyanobacteria (Fig. 2) with lower contributions from dinophytes, while euglenophytes were rare. Diazotrophic cyanobacteria, of which *Dolichospermum sp.* was the main taxon (we found only a few colonies of *Anabaenopsis tanganyikae* colonies in the South), were most abundant in the North and centre of the lake, where the primary thermocline was below the euphotic zone (Apr/May only) or deeper than ~40 m (Fig. 2 & 4). This observation is supported by the breakpoint analysis showing a threshold depth in the relationship

between the relative abundance of diazotrophs and primary thermocline at 36.7 m (excepting station 9 in Sep/Oct and station 7 in Apr/May). We observed the highest *Dolichospermum* as well as total phytoplankton abundances (1.2 · 10$^9$ colonies m$^{-2}$ and 3.3 · 10$^9$ ind. m$^{-2}$, respectively) at station 3 during Apr/May. *Dolichospermum* colonies consisted on average of 36 and 44 vegetative cells and ca. 2-3 heterocysts in Sep/Oct and Apr/May, respectively (Fig. 5). The DIN-assimilating cyanobacteria *Chroococcus sp.* were particularly abundant in the very North as compared to the South. The diatoms *Nitzschia* were most

abundant in the North and centre, but rare in the South (stations 7-9) where other diatoms of the genera *Navicula* and *Cymbella* were abundant in Apr/May. In Sep/Oct, when diatoms were generally less abundant in the South, dinophytes reached their maximum relative abundances (up to 24 %).

While the absolute abundances of the total phytoplankton community were similar during the two expeditions (0.6-3.3 · 10$^9$

ind. m$^{-2}$), cyanobacteria and diatoms reached higher relative abundances in Apr/May compared to Sep/Oct at the expense of chlorophytes. Especially the genera *Dolichospermum* and *Nitzschia* were thriving in Apr/May, usually bound to the surface between 0 and 25 m (Fig. 2 & 3). The abundances of those two genera were significantly correlated with each other in Apr/May (p < 0.001, Pearson correlation coefficient: 0.62). Often we found single *Nitzschia* cells or entire clusters entangled in *Dolichospermum* colonies (Fig. 5). Less frequently, other taxa such as *Oocystis* and other chlorophytes were apparently

associated to *Dolichospermum* colonies.

### 4 Discussion

### 4.1 Biogeochemical bottom-up control of diazoptroph abundance

Our data imply that high densities of diazotrophs (esp. *Dolichospermum*) formed in response to reduced DIN fluxes into the surface waters in Lake Tanganyika (Fig. 1). A breakpoint analysis supports the hypothesis that the location of the primary

thermocline (Fig. 4a) and its relative position to the euphotic depth (Fig. 4b) play a role for creating a niche for diazotrophs by





controlling the upward DIN transport into the euphotic zone. The modelled threshold depth of 37 m corresponds well to previous estimates of the mean euphotic zone thickness in Lake Tanganyika, ranging mostly between 30 and 40 m (Cocquyt and Vyverman, 2005; Descy et al., 2005). In congruence with the substantial numbers of diazotrophs in Sep/Oct, when the primary thermocline was located below ~40 m, the euphotic depth on average reaches down to ~40 m during that time (Cocquyt

and Vyverman, 2005). This is shallower than in the less productive rainy season (Cocquyt and Vyverman, 2005), where diazotrophs have been found to accumulate only where the primary thermocline was located below 45 m (Fig. 4a).

Station 8 in Apr/May provides a striking example of a scenario where the sufficient N supply to the euphotic zone controls the phytoplankton community structure (Fig. 3b). With the thermocline located well within the euphotic zone, vertical $NO_3^-$ fluxes

were efficient and the phytoplankton community was dominated by diatoms (esp. *Navicula*, *Cymbella*, and *Nitzschia)* and chlorophytes. These DIN-assimilating taxa preferably inhabited the lower region of the euphotic zone (Fig. 3b) where levels of light and nutrients were both optimal. Consequently, the Chl peak was situated at the bottom of the euphotic zone and PE and PC concentrations were low.

In contrast to the observations from station 8, a deep thermocline restricting the vertical $NO_3^-$ flux into the euphotic zone was found at stations 2 to 5 during Apr/May (Fig. 1, 3a, & S4). Here, the primary thermocline was situated well below the euphotic zone and PE and PC concentrations as well as diazotroph abundances were high. The limited $NO_3^-$ supply created a niche for diazotrophs and their potential partners near the surface resulting in a shallow Chl peak (also see Brentrup et al., 2016 for vertical shifts of Chl peak). In accordance, surface $NO_3^-$ was often below detection limit (Fig. 1), whereas $PO_4^{3-}$ was available

in excess at all stations and during both campaigns (Fig. S2). Along with diazotrophs, diatoms and chlorophytes were the most important phytoplankton taxa, confirming the typical rainy season cyanobacteria-chlorophytes-diatom community described by Cocquyt and Vyverman (2005).

The behaviour of the two outliers in the breakpoint analysis (Fig. 4) deserves some further consideration. For example, at

station 9 in Sep/Oct we observed substantial diazotroph abundances ($8.5 \cdot 10^7$ colonies m$^{-2}$) despite weak stratification (Table S3) and relatively high surface $NO_3^-$ concentrations (Fig. 1 & 2). In this case, the very high N deficit of the upwelled water (Fig. S2a) might have stimulated N fixation in analogy to a nutrient pulse in coastal systems (Vahtera et al., 2007). The persistent emergence of high *Dolichospermum* numbers at the southern end of the lake (Langenberg et al., 2002; Vuorio et al., 2003) implies that they might also be able to thrive in waters where absolute DIN concentrations are relatively high. The

southernmost stations (8 & 9) were characterized by extremely high amounts of excess $PO_4^{3-}$ (DIN:$PO_4^{3-}$ < 4:1) likely originating from mixing with intermediate waters with DIN:$PO_4^{3-}$ ratios well below 5:1 (Edmond et al., 1993). The dynamics of those recurring events, including the effects of rapidly warming surface water as well as the high N deficit, require further examination. The absence of diazotrophs at station 7 (Apr/May) can be attributed to a lagged response of the phytoplankton community to $NO_3^-$ deficiency, because the thermocline is located just below the euphotic zone. For further information





regarding the local NO$_3^-$ peak at station 3, potential inflow from the Malagarasi river as well as the thermal structure of the lake see supporting information sections 2 and 5.

## 4.2 Additional control via DIN regeneration and phytoplankton interactions

A critical analysis of the NO$_3^-$ data and the diazotroph abundances reveals an apparent controversy: In cases where the primary
thermocline was located within the euphotic zone, the surface zone was often virtually nitrate-exhausted but diazotrophs remained scarce. From a physical perspective, such a shallow thermocline will retain sinking particles within the euphotic zone and thus enhance remineralization, providing freshly regenerated DIN for DIN-assimilating taxa. Moreover, diazotrophs might be Fe-limited when DIN-assimilating phytoplankton thrive. Despite high atmospheric Fe inputs in the savannah-like landscape some authors claimed that due to the high pH of the surface water, bioavailability of Fe may be low for some phytoplankton
taxa (North et al., 2008; De Wever et al., 2008). Enhanced activity of DIN-assimilating taxa will further exhaust Fe that would otherwise be available for the Fe-expensive N fixation (Maldonado and Price, 1996). An additional ecological driver for diazotroph abundance may operate by mutualistic interactions between diazotrophs and diatoms (Foster and O'Mullan, 2008; Foster and Zehr, 2019).

While a mutualistic association or *in situ* growth together with diatoms might be the best way for diazotrophs to access bioavailable Fe under DIN-exhausted conditions in the euphotic zone, there might be no selection for such an interaction under the presence of DIN. Associations between diazotrophs and diatoms were previously reported in Lake Tanganyika (Van Meel, 1954) and our data shows that when diazotrophs co-occurred with *Nitzschia* they appeared also to be physically attached to one another (Fig. 5). Furthermore, high abundances of diazotrophs were associated with a surface Chl maximum, indicating
that both cyanobacteria and their potential partners were the key players in the phytoplankton community (Fig. 1). Within these potential associations, diazotrophic colonies contained numerous heterocysts allowing for efficient N fixation. Single cell techniques revealed that diazotrophs fix up to 420 times more N in the presence of their diatom partner highlighting the importance of the symbiosis for high N fixation rates (Foster et al., 2011). The exact role of the diatoms is still uncertain, and may include the provision of readily available carbon (Harke et al., 2019) and iron, which they can extremely efficiently
scavenge (Kazamia et al., 2018), while benefiting from up to 97.3 % of the fixed N (Foster et al., 2011).

## 4.3 Conceptual scenarios explaining diazotroph dominance

We propose a simple observation-based concept for identifying potential niches for diazotrophs in Lake Tanganyika. The emergence of these niches are characterized by the location of the thermocline and the euphotic depth as well as potential
synergistic interactions within the phytoplankton community (Fig. 6). Our data support the assumption that primary productivity is N limited and that micronutrients needed for N fixation are available. The formation of diazotroph niches are defined by two scenarios:



*1) thermocline within the euphotic zone*

In this scenario, the vertical transport of $NO_3^-$ into the euphotic zone is rapid. The high light and nutrient availability allow DIN-assimilating phytoplankton to thrive. Hence, there is no environmental selection for potentially mutualistic interactions between *Nitzschia* and diazotrophs. The Chl peak is located in the lower part of the euphotic zone, where both light and nutrients are available, whereas diazotroph abundances are low.

*2) thermocline below the euphotic zone*

Here, the thermocline severely slows down the vertical transport of $NO_3^-$ into the euphotic zone. The poor availability of $NO_3^-$ results in a lack of N for DIN-assimilating phytoplankton. This creates a favourable environment for diazotrophs and their potential partners near the water surface. Hence, diazotrophs and *Nitzschia* are the key players under such conditions and the Chl peak is located near the surface.


Our concept explains the diazotroph abundance from a bottom-up perspective, but an alternative explanation could include effects stimulating selective retention of diazotrophs. While horizontal transport might be responsible for some of the observed variability, it could also result in accumulation of positively buoyant cyanobacterial cells. Moreover, biological processes such as preferential zooplankton grazing certainly can play a role. Even if selective retention is a significant factor, diazotrophs first

needed to find suitable conditions to grow before being accumulated by physical or ecological processes. Systematically investigating the role of interval waves in Lake Tanganyika for establishing or disrupting the conditions outlined in our concept with time series data is an important next step for further constraining the factors regulating diazotroph abundance in this lake. To reduce effects of internal waves, both our field campaigns were scheduled when thermocline oscillations should have been relatively minor, i.e. Sep/Oct, when the thermocline just re-stabilizes and Apr/May, when internal waves already lost energy

over the course of the entire calm rainy season. The general applicability of our concept is limited by the snapshot sampling, but recent modelling (Follett et al., 2018) and previous field studies (Cocquyt and Vyverman, 2005; Dore et al., 2008; Post, 2005; White et al., 2007) support the idea that reduced supply of DIN into the surface zone during periods of strong stratification can provide favorable conditions for diazotrophic cyanobacteria in oligotrophic waters.

## 5 Conclusions

Our study provides field evidence biogeochemical drivers regulating the abundance of diazotrophic cyanobacteria in Lake Tanganyika during the seasonal transition in October and the calm rainy season. The conceptual scenarios offer a plausible, mechanistic perspective on potential factors controlling the proliferation of diazotrophic cyanobacteria in N-deficient, oligotrophic waters. Testing the validity of this concept in comparable lakes or ocean regions is logistically easy, because measuring the thermocline location and euphotic depth as well as diazotroph abundance directly or via proxies is already

included in many monitoring programs. Our results also show that the fluorometric determination of extracted phycocyanin

**Biogeosciences** Open Access
Discussions
EGU

and phycoerythrin provides an excellent proxy for the abundance of filamentous cyanobacteria. Additional ecological drivers of diazotroph dominance, however, still remain elusive. Hence, detailed studies focussing on the interactions between diazotrophs and other phytoplankton taxa are needed. As global warming induces stronger stratification in parts of the ocean and lakes worldwide, the ecological implications of reduced vertical mixing on aquatic ecosystems become an urgent issue.

**Data availability**

Data and metadata are available in the ETH Zurich Research Collection repository (Ehrenfels et al., 2020).

**Author contribution**

BE and BW designed the study. BE, ASM, KBLB, CD, JJ, TK, IAK, and EAS organized and contributed to field work. BE, ASM, KBLB, CD, JJ, and TK performed field and laboratory measurements. BE, ASM, and DO analyzed data. BE
conceptualized ideas for the paper and wrote the manuscript with input from all co-authors. MB, FP, and BW contributed to interpretation and development of the manuscript.

**Competing interest**

The authors declare that they have no conflict of interest.

**Acknowledgements**

This work would not have been possible without the efforts of our research collaborators from the Tanzania Fisheries Research Institute, particularly the Directors Rashid Tamatamah and Semvua Mzighani as well as Mary Kishe. Asante sana. We are grateful to Anthony Kalangali for support in the field and lab. We owe special thanks to Mupape Mukuli as well as the captain and crew of the M/V *Maman Benita* for their organizational efforts and technical support during the sampling campaigns. We also thank Andreas Brand for field assistance, the Aquatic Physics Group at Eawag for their help with interpreting the
hydrodynamics of Lake Tanganyika, Patrick Kathriner, Marta Reyes, and Daniel Steiner for lab assistance, and Eliane Scharmin, Luzia Fuchs, and Patricia Achleitner for their administrative support. This project was funded by the Swiss National Science Foundation (grant 166589).

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



## Figures

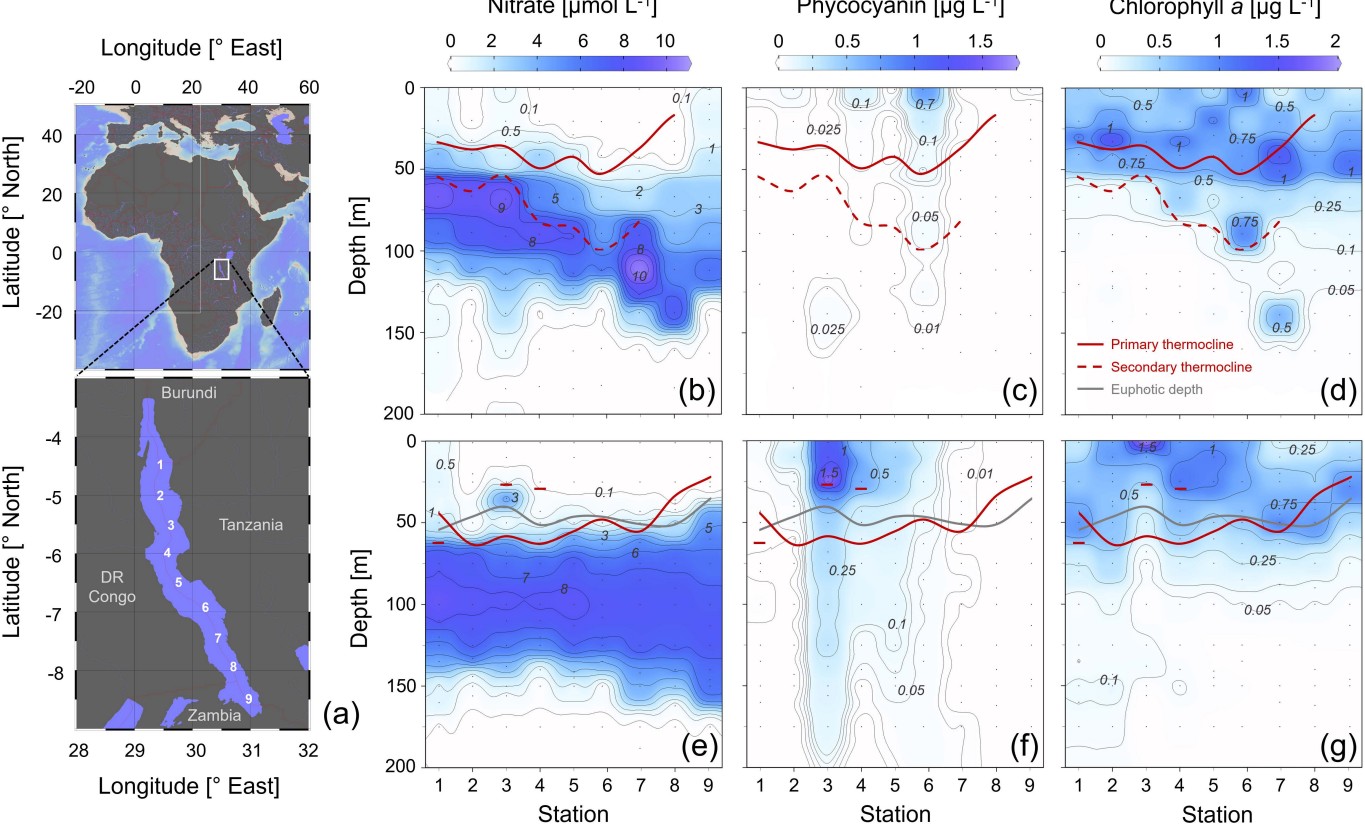

**Figure 1:** Map of Lake Tanganyika showing the 9 sampling stations in a North-South transect **(a)**. Distribution of the nitrate **(b,e)**, phycocyanin **(c,f)**, and chlorophyll *a* concentrations **(d,g)** in Lake Tanganyika including the locations of the primary thermocline and the euphotic depth as well as shallow and deep secondary thermoclines. The upper panels correspond to the end of the cool dry season (September/October 2017) and the lower panels to the end of the more stratified, warm rainy season (April/May 2018). Note that surface nitrate concentrations below 0.4 µmol/L were often below detection limit. Dots indicate samples.






**Figure 2:** The depth integrated phytoplankton abundances (top) and community compositions (bottom) at the 9 sampling stations during **(a)**
the end of the dry season (September/October 2017) and **(b)** the end of the rainy season (April/May 2018). Note that the depth of taxonomic
hierarchy varies to highlight functionally relevant taxa. We divided diatoms into the genus *Nitzschia* and others. Cyanobacteria were
partitioned by genus with the exception of *Dolichospermum* and *Anabaenopsis*. We pooled these heterocystous and filamentous genera due
to their capability of fixing atmospheric nitrogen (diazotrophs).



**Figure 3:** Vertical profiles of physical, biogeochemical and biological data from two exemplary stations at the end of the rainy season (April/May 2018). Station 5 **(a)** represents a thermocline below the euphotic zone, whereas station 8 **(b)** exemplifies a thermocline within the euphotic zone. The left panels show temperature (T), euphotic depth ($z_{eu}$), nitrate ($NO_3^-$), phosphate ($PO_4^{3-}$), chlorophyll (Chl), and phycocyanin (PC), whereas the right panels represent the abundances of different members of the phytoplankton community. Note that the depth of taxonomic hierarchy varies to highlight functionally relevant taxa.

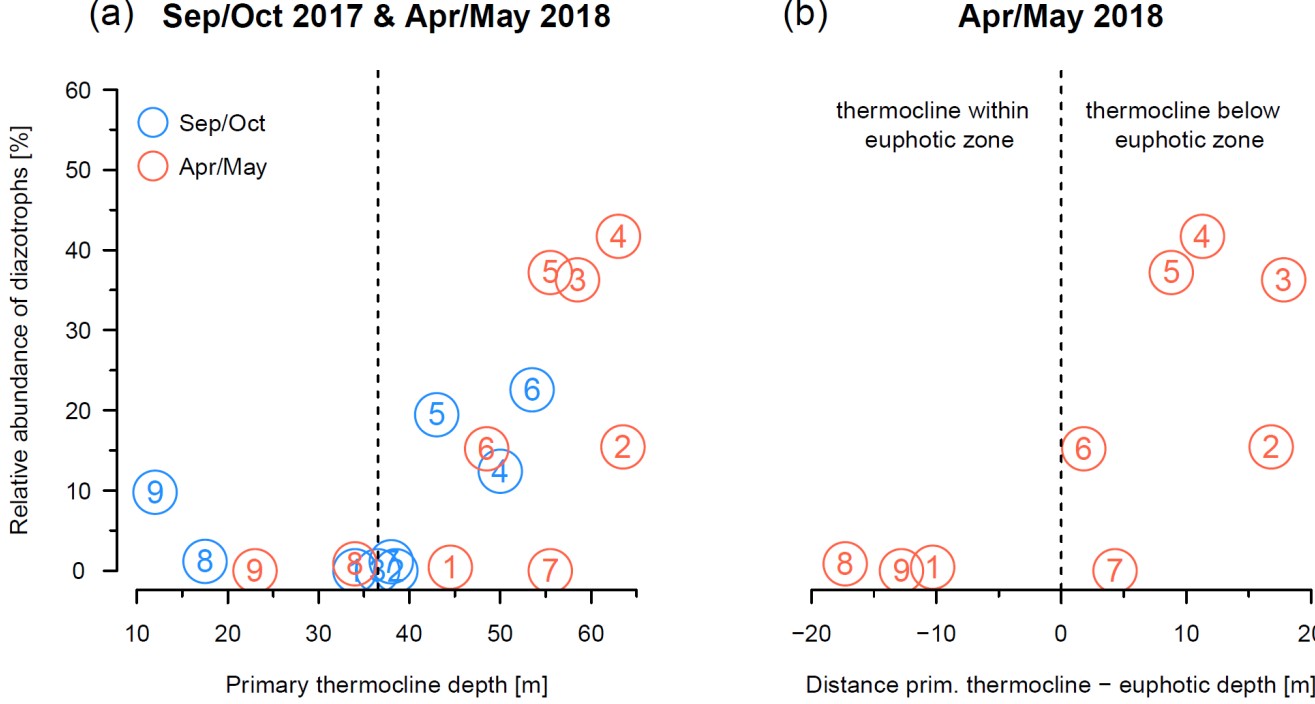

**Figure 4: (a)** The depth integrated relative abundance of diazotrophs for the 9 sampling stations as a function of primary thermocline depth for the end of the dry season (September/October 2017) and the end of the rainy season (April/May 2018) and **(b)** the distance between euphotic depth and primary thermocline location for the warm season (irradiance data for Sep/Oct missing). In Sep/Oct, the relative diazotroph abundances were high if the thermocline was located below 40 m. A breakpoint analysis identified 36.7 m as breakpoint location in the relationship between these two variables. In Apr/May, the relative abundances of diazotrophs were high if the thermocline was located below the euphotic zone.



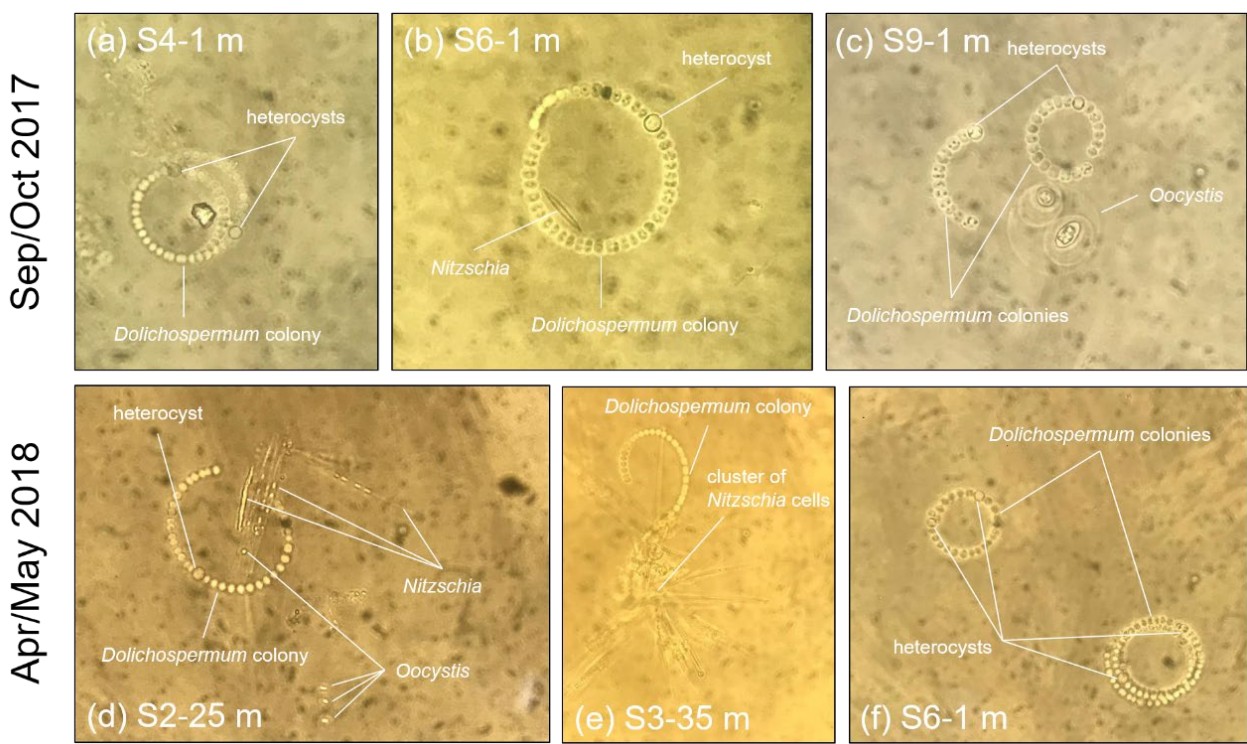

**Figure 5:** Microscopic images of *Dolichospermum* colonies with heterocysts and potential associations with other taxa at various sites during September/October 2017 (top) and April/May 2018 (bottom). Panels **(a)** and **(f)** show *Dolichospermum* colonies in the absence of other taxa. Panels **(b)** and **(c)** depict *Nitzschia* and *Oocystis* cells in close vicinity of *Dolichospermum* colonies, whereas **(d)** and **(e)** reveal clusters of *Nitzschia* apparently being entangled in *Dolichospermum* colonies.



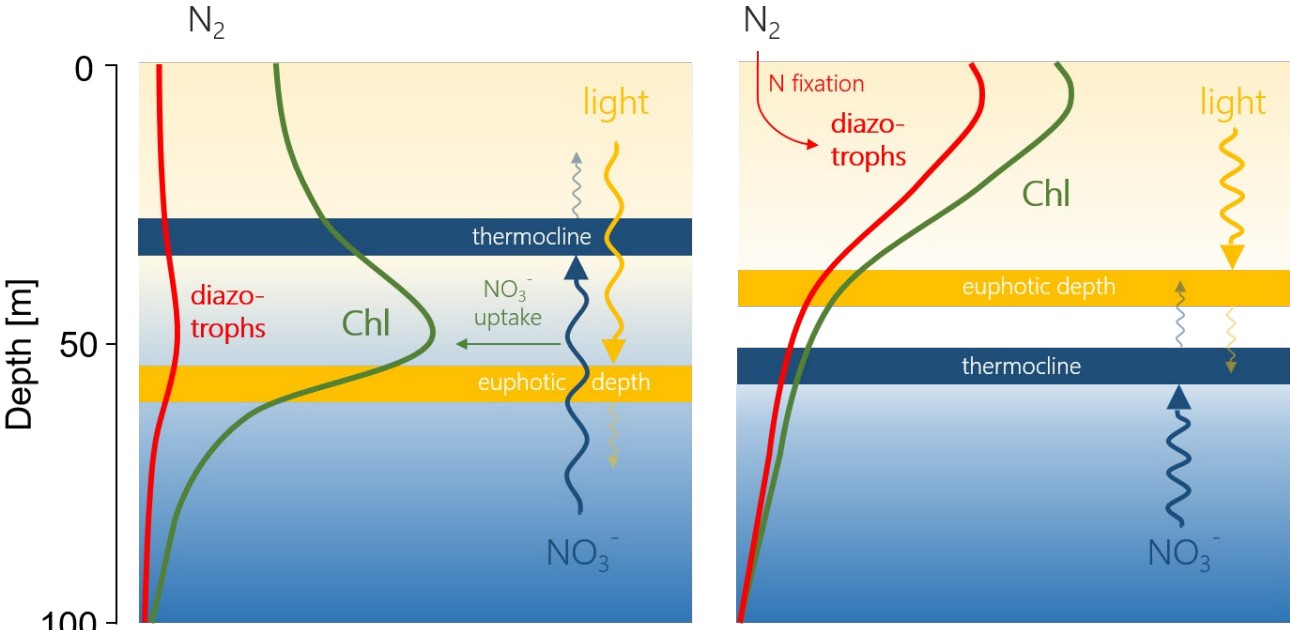

**Figure 6:** Two conceptual scenarios explaining the diazotroph abundance and chlorophyll a (Chl) concentrations as response to the thermocline location and the euphotic depth in Lake Tanganyika. **(a)** The thermocline is located within the euphotic zone. Therefore, the flux of deep-water nitrate ($NO_3^-$) into the euphotic zone is high. Dissolved inorganic nitrogen (DIN) assimilating phytoplankton outcompete diazotrophs and form the subsurface Chl peak, because they are the key players under the high light and nutrient availability in the lower part of the euphotic zone. **(b)** The thermocline is located below the euphotic zone. Hence, the reduced flux of deep-water $NO_3^-$ into the euphotic zone creates a favourable environment for mutualistic interactions between diazotrophs and diatoms, because they can exploit atmospheric nitrogen ($N_2$) as a N source while other DIN-assimilating phytoplankton have to rely on $NO_3^-$. Diazotrophs and their potential partners are highly abundant near the water surface resulting in a near surface Chl peak.