# Peer review of "Thermocline depth and euphotic zone thickness regulate the abundance of diazotrophic cyanobacteria in Lake Tanganyika"

_Biogeosciences, 2020_

## Referee Comment (RC1) · Anonymous Referee #1 · 30 Jun 2020

General comments The manuscript presents a conceptual model for the development heterocytous cyanobacteria in Lake Tanganyika, involving the position of the thermocline and euphotic depth. The data were acquired during two cruises on the lake, carried out at the end of the rainy season and at the end of the dry season. The model is presented as possibly explaining "blooms" of cyanobacteria in N-deficient surface water systems. However, the authors apparently ignored key aspects of the present phytoplankton assemblage of Lake Tanganyika and some claims are not based on sufficient data (such as N vs. P limitation). Specific comments Regarding the methods, those for physical and chemical analyses are correct and yielded apparently consistent data. On the contrary, the phytoplankton analyses, being based on concentrating the

phytoplankton on a 10 $\mu$m plankton net could not give a correct sample for quantitative phytoplankton analysis: Lake Tanganyika comprises taxa in a large size range, with a substantial, often dominant part of nanoplankton < 10 $\mu$m and of picoplankton (< 2 $\mu$m). Even though in the methods it is clear that this sampling method focused on medium to large sized phytoplankton, in the results, such a statement as "The phytoplankton community in Lake Tanganyika was dominated by chlorophytes, diatoms, and cyanobacteria . . ." is misleading. See also below the remark about the correlation of PC and PE data with cyanobacteria biomass. In addition to this serious technical problem, there are two main issues in this study, in addition to various potential shortcomings (see other remarks below). The first is the assumption (lines 52-53) that "cyanobacteria can dominate the phytoplankton community during periods of pronounced stratification when nutrients are scarce (Cocquyt and Vyverman, 2005)". This is not untrue, but the cyanobacteria that dominates the phytoplankton in present Lake Tanganyika are not the heterocytous taxa but picocyanobacteria (Synechococcus spp.), which have quite different characteristics. Briefly, they are not as efficient N-fixers as the heterocytous taxa, and their small size allows high growth rates and high nutrient uptake rates, which make them specialists of oligotrophic conditions. This is an issue throughout the manuscript, which does not mention these picocyanobacteria, which make 41 - 99 % of total phytoplankton biomass (Stenuite et al., JPR, 2009). The paper by Cocquyt & Vyverman, by contrast, being based only on LM examination, does not account for "algae" < 5 $\mu$m, and therefore gives a biased view of the phytoplankton assemblage of the present lake. Actually, the heterocytous taxa (mainly Dolichospermum, formerly Anabaena) are presently detected relatively rarely in Lake Tanganyika, as shown by analyses of samples collected over a few years (Descy et al., Hydrobiologia, 2010), as well as by remote sensing, which allowed to detect surface "blooms" (Horion et al., 2010). It seems that they occur much less frequently than in the past, which may be a consequence of the lake's oligotrophication (Verburg et al., 2003). A second issue relates to the assumption that N is the main limiting factor of phytoplankton growth in the lake (line 72-73), which is in contradiction with evidence based on seston elemental

ratios (which is a more reliable indicator of phytoplankton nutrient status than a deficit estimated from a ratio DIN : SRP). According to Stenuite et al. (2007), who used particulate C, N and P analyses to assess nutrient limitation over several years in 2 sites in Lake Tanganhyika, P limitation was more frequent than N-limitation, and neither nutrient could be considered as severely limiting. Both statements convey the impression that N-fixation is a key process controlling productivity in the present lake, whereas there is evidence that, as a result of global warming and increased stratification, the lake's productivity has decreased as a result of decreased P availability (Verburg et al., 2003, 2006), with consequences on fish yield, although this is still a matter of debate (Verburg et al., 2006; Sarvala et al., 2006). Interestingly, the increased P availability may have been the cause for the reported decline of heterocytous cyanobacteria, as their P requirements are typically high. An alternative conceptual model for Dolichospermum may be different from the one proposed in the paper: indeed, several authors have emphasized that the typical timing of surface "blooms" in Lake Tanganyika is the transition between the dry and the rainy season, in October-November, when they have exploited the SRP-rich conditions of the dry season, possibly by storing polyphosphate granules; only when the lake re-stratifies, they can outcompete the other phytoplankton, at least for a time, using the advantages of buoyancy due to their gas vesicles and of efficient N fixation with their heterocytes. "Blooms" occurring at other times, as in the moderately nutrient-limiting conditions of the middle and end of the rainy season, may be explained by migration below the thermocline to take up and accumulate SRP. Low Fe availability at high pH may be an additional controlling factor of N-fixation, as suggested by experimental nutrient additions in tropical lakes, and grazing resistance may be an additional factor of success of these large-sized phytoplankters. An interesting analogue to heterocytous cyanobacteria development in an oligotrophic tropical lake can be found with the case of Trichodesmium in the ocean, which has been given a lot of attention. The conceptual model for Trichodesmium also involves N-fixation (with diazocytes, not heterocytes), control of growth by SRP and Fe, and vertical migration regulated by light and nutrient requirements (Bergman et al., 2012).

Other remarks 73-74: "This large tropical lake is permanently stratified and characterized by a nutrient-rich hypolimnion overlaid by an oligotrophic epilimnion" Poor depiction of the lake ... better to mention that it is meromictic, and that the mixolimnion is stratified during the wet season and mixed during the dry season, with a large spatial variability.

106: "This sampling approach has been adopted in previous studies (e.g. Salonen et al., 1999; Stenuite et al., 2009)": Yes but long time series with adequate sampling frequency are better suited to capture temporal variability, particularly when studying relatively rare "bloom" events. Combination with remote sensing data is ideal, given the very large spatial heterogeneity (see Horion et al. 2010) 135: "We calculated the N deficit according to the Redfield stoichiometry of phytoplankton ..." Very rough indeed, as it depicts only the situation at the time of sampling; moreover, a concentration ratio is not a supply ratio, and only nutrient supply vs. demand from phytoplankton determines nutrient status 178 : "The N deficit (98 % of all observations) persisting throughout the water column (Fig. S2) implies that primary productivity was N limited": far from sure, see above 205: "The phytoplankton community in Lake Tanganyika was dominated by chlorophytes, diatoms, and cyanobacteria (Fig. 2) with lower contributions from dinophytes, while euglenophytes were rare. Diazotrophic cyanobacteria, of which Dolichospermum sp. was the main taxon (we found only a few colonies of Anabaenopsis tanganyikae colonies in the South), were most abundant in the North and centre of the lake, where the primary thermocline was below the euphotic zone (Apr/May only) or deeper than ∼40 m (Fig. 2 & 4)". Again, the statement implies that cyanobacteria were all efficient diazotrophs, which was likely not the case. Cryptophytes were overlooked. 276-277 "An additional ecological driver for diazotroph abundance may operate by mutualistic interactions between diazotrophs and diatoms" and below (285-290). This rather wild assumption is just based on "contact" between cyanos and diatoms, which may have resulted from the sampling procedure. This has nothing to do with the endosymbiosis of Richelia with large diatoms in the sea. 330: "Our results also show that the fluorometric determination of extracted phycocyanin and phycoerythrin provides

an excellent proxy for the abundance of filamentous cyanobacteria" This statement is plainly false, as both phycobilins are also present in picocyanobacteria, which dominate the phytoplankton in the present lake. The fact that PC and PE concentration correlated with filamentous cyanobacteria abundance was very likely pure chance. Table S2 contains several misspelling of taxa names, likely some wrong identifications and a mis-classification (Sphinctosiphon is a cyanoprokaryote, not a green alga).

---

## Author Comment (AC1) · 16 Jul 2020

We thank the Reviewer very much for the thorough review of our work. We are grateful for all the time and effort that the Reviewer has invested in providing constructive criticism. We have considered all comments and are convinced that the elaborate feedback will help us improve our manuscript.

We have addressed the comments (blue) with a point-by-point reply (black).

**Comment:**

Even though in the methods it is clear that this sampling method focused on medium to large sized phytoplankton, in the results, such a statement as "*The phytoplankton community in Lake Tanganyika was dominated by chlorophytes, diatoms, and cyanobacteria…*" is misleading.

**Reply:**

We understand that the chosen phrasing in the results section might invite interpretations about the entire (and not only the investigated part of the) phytoplankton community. The purpose was not, however, to mislead readers but rather to provide an overview of the community within the sampled phytoplankton size fraction.

**Anticipated changes:** In the revised version, we will make it very clear that the entire study is focused on microscopic (>10 µm) phytoplankton and that we cannot make any strong assertions about either smaller fractions or about the total phytoplankton biomass in the lake.

**Comment:**

"*cyanobacteria can dominate the phytoplankton community during periods of pronounced stratification when nutrients are scarce (Cocquyt and Vyverman, 2005)*".

This is not untrue, but the cyanobacteria that dominates the phytoplankton in present Lake Tanganyika are not the heterocytous taxa but picocyanobacteria (Synechococcus spp.), which have quite different characteristics.

**Reply:**

We agree that the potential importance of picocyanobacteria needs to be addressed in the revised version of the manuscript. Our study, however, focused on the well-documented main N fixing cyanobacteria (filamentous genera *Dolichospermum* and *Anabaenopsis*), which as indicated by the Reviewer, are ecologically different and may therefore also be favored by different environmental conditions. In our study, we find the filamentous N fixers to be very successful when the surface zone is DIN-exhausted and the upward transport of metalimnetic DIN is slowed down by a deep thermocline. As discussed in more detail below, this is quite contrary to the high nutrient situations under which *Synechococcus* has been found to thrive in Lake Tanganyika (e.g. Descy et al., 2005 & 2010).

The aim of this particular sentence the Reviewer is referring to was, however, to provide an introduction and context and we were describing a specific situation (nutrient-scarce, stratified conditions) under which the large diazotrophic cyanobacteria can indeed reach high densities in the lake (Cocquyt and Vyverman, 2005; Descy et al., 2010; Hecky and Kling, 1981; Salonen et al., 1999). Moreover, several studies have reported that both picocyanobacteria and filamentous taxa can dominate the cyanobacterial community in Lake Tanganyika (see e.g. Descy et al., 2005, 2010; Salonen et al., 1999; as well as the quotes below).

**Anticipated changes:**

Replying to some of the more specific comments and remarks below, we have indicated how and in what context the importance of picocyanobacteria will be discussed in the next manuscript version. Furthermore, we will clearly highlight that we focus on the filamenous, diazotrophic cyanobacteria.

**Comment:**

the heterocytous taxa (mainly Dolichospermum, formerly Anabaena) are presently detected relatively rarely in Lake Tanganyika, as shown by analyses of samples collected over a few years (Descy et al., Hydrobiologia, 2010), as well as by remote

sensing, which allowed to detect surface "blooms" (Horion et al., 2010). It seems that they occur much less frequently than in the past

**Reply:**

We agree that the scenarios we observed during our two lake-wide surveys may not be the most common and only occur sporadically in the lake. In our case study, we have thus made no claims about the extent of the outlined mechanistic.

Nonetheless, we have found no solid evidence supporting the long-term decline in blooms of filamentous cyanobacteria in Lake Tanganyika and would greatly appreciate if the Reviewer could share the sources of that statement.

Descy et al. (2010) write: "*The dominant cyanobacteria were coccal unicells and colonies (mostly Synechococcus sp., Aphanocapsa spp., Gloeothece hindakii and Chroococcus spp.), or filaments (mainly Anabaena spp. [= Dolichospermum] and Anabaenopsis tanganyikae).*"

Similarly, Horion et al. (2010) state: "*Anabaena sp. [= Dolichospermum] and other filamentous cyanobacteria (such as Anabaenopsis tanganyicae) are N-fixers, which allows them to outcompete other phytoplankton in DIN-depleted environments, typical of the surface waters of Lake Tanganyika during the rainy season.*"; further supporting the observations from previous studies showing the importance of filamentous cyanobacteria during warming period at the dry-rainy season transition (Cocquyt and Vyverman, 2005; Hecky and Kling, 1981; Langenberg et al., 2002; Narita et al., 1986) and strong stratification during the rainy season (Descy et al., 2005; Salonen et al., 1999; Vuorio et al., 2003).

**Comment:**

A second issue relates to the assumption that N is the main limiting factor of phytoplankton growth in the lake (line 72-73), which is in contradiction with evidence based on seston elemental ratios (which is a more reliable indicator of phytoplankton nutrient status than a deficit estimated from a ratio DIN : SRP). According to Stenuite et al. (2007), who used particulate C, N and P analyses to assess nutrient limitation over several years in 2 sites in Lake Tanganhyika, P limitation was more frequent than N-limitation, and neither nutrient could be considered as severely limiting.

135: "*We calculated the N deficit according to the Redfield stoichiometry of phytoplankton : : :*" Very rough indeed, as it depicts only the situation at the time of

sampling; moreover, a concentration ratio is not a supply ratio, and only nutrient supply vs. demand from phytoplankton determines nutrient status 178 : "*The N deficit (98 % of all observations) persisting throughout the water column (Fig. S2) implies that primary productivity was N limited*": far from sure, see above

**Reply:**

Agreed. Dissolved nutrient ratios do not necessarily provide a complete picture into the nutrient status of the phytoplankton. What was indicative for the N limitation is not only the relative N deficit, but also the fact that at 8 out of 9 stations (Apr/May) and 6 out of 9 stations (Sep/Oct), the levels of free nitrate in the surface waters were below the detection limit and more often so when the thermocline was below the euphotic zone (i.e., the supply of nitrate from deeper waters was low).

Additionally, Stenuite et al. (2007) also suggested that N limitation occurred during their study : "*Both C : P and C : N ratios suggest that macronutrient limitation of phytoplankton may have occurred in Lake Tanganyika…, but also that P limitation may have been more frequent than N limitation.*". It is worth considering that the abovementioned estimates, based on Redfield stoichiometry and sole analysis of seston elemental ratios, may also entail some potential issues i.e., bias from allochthonous material or heterotrophic biomass.

**Anticipated changes:**

Nonetheless and closely following this comment, we have analyzed and will present now (in the revised version) data on the seston N:P ratios that, according to Stenuite et al. (2007), are indicative of phytoplankton N limitation during our surveys. In addition, we will more elaborately explain why the phytoplankton was N limited during our surveys including the remarks above.

**Comment:**

Both statements convey the impression that N-fixation is a key process controlling productivity in the present lake, whereas there is evidence that, as a result of global warming and increased stratification, the lake's productivity has decreased as a result of decreased P availability (Verburg et al., 2003, 2006), with consequences on fish yield, although this is still a matter of debate (Verburg et al., 2006; Sarvala et al., 2006). Interestingly, the increased[decreased?] P availability may have been the cause

for the reported decline of heterocytous cyanobacteria, as their P requirements are typically high.

**Reply:**

We agree with the Reviewer, that we cannot know to what extent N fixers control the overall productivity in Lake Tanganyika. Nevertheless, many studies have concluded that N fixation might be very important for the nutrient budget in a lake with such low dissolved N:P ratios (Bootsma and Hecky, 1993; Hecky, R. E., Spigel, R. H., & Coulter, 1991; Hecky et al., 1996). We do not doubt that the climate change-induced increase in stratification can reduce primary productivity in Lake Tanganyika, but it is not clear to us why less vertical mixing would only decrease the availability of P and not DIN, as implied by the Reviewer. Furthermore, in the studies cited by the Reviewer we did not find a clear explanation why P would be the primary control on overall primary productivity, especially if regarded in the context of a highly unbalanced dissolved N:P budget. We can only assume that the authors of those studies followed the notion from the limnological studies by Hecky et al. that N fixation will compensate (at least partially) the deficit in dissolved N compared to P.

Despite these open questions, N fixation has not been investigated in Lake Tanganyika. In our study, the abundant presence of heterocycsts within the *Dolichospermum* colonies clearly show that they were fixing N. Moreover, our $^{15\text{-}15}N_2$ incubation experiments conducted during the same cruises revealed high N fixation rates (we will present those in a separate publication), which follow the pattern of diazotroph abundance. We are therefore convinced that N fixation can be an important process in Lake Tanganyika at least for the studied periods.

**Anticipated changes:**

We will make sure, however, to revise the text in order to remove any implications that N fixers are controlling the overall productivity of the lake.

**Comment:**

An alternative conceptual model for Dolichospermum may be different from the one proposed in the paper: indeed, several authors have emphasized that the typical timing of surface "blooms" in Lake Tanganyika is the transition between the dry and the rainy season, in October-November, when they have exploited the SRP-rich conditions of the dry season, possibly by storing polyphosphate granules; only when

the lake re-stratifies, they can outcompete the other phytoplankton, at least for a time, using the advantages of buoyancy due to their gas vesicles and of efficient N fixation with their heterocytes.

**Reply:**

Our surveys were timed for periods typical for *Dolichospermum* blooms, both during the end of the dry season (Cocquyt and Vyverman, 2005; Hecky and Kling, 1981; Langenberg et al., 2002; Narita et al., 1986) as well as during the rainy season (Descy et al., 2005; Salonen et al., 1999; Vuorio et al., 2003).

We appreciate the interesting points made by the Reviewer, but do not think that they are in contradiction to our concept. On the contrary, this neatly places the basic idea of the paper (diazotrophs have an advantage when N is scarce, but P available) into the temporal context of the transition from dry to rainy season. However, during our sampling we have neither observed DIN- or SRP-exhausted conditions in the south at the end of the dry season (see station 9, Sep/Oct 2017; data file "samples_sep-oct_2017.csv", columns "PO4" and "NO3" measured according to Grasshoff et al., 1999 and Schnetger and Lehners, 2014). The suggested mechanism might be more relevant just after the surface nutrients are depleted.

**Anticipated changes:**

We are expanding the discussion to include this possible temporal development beyond our study period.

**Comment:**

 "Blooms" occurring at other times, as in the moderately nutrient-limiting conditions of the middle and end of the rainy season, may be explained by migration below the thermocline to take up and accumulate SRP.

**Reply:**

We do not disagree that there might be another complementary mechanism stimulating diazotrophic cyanobacteria. Notwithstanding given our extensive biogeochemical dataset that is fully accessible as a Supplement to this manuscript we do not grasp the immediate importance of the suggested mechanism here. Why would the visually counted *Dolichospermum* need to rely on vertical migration for P uptake, if there is still sufficient SRP to support its growth (see data files "samples_sepoct_2017.csv" and "samples_apr-may_2018.csv", columns "PO4" measured according to Grasshoff et al., 1999)?

**Comment:**

Low Fe availability at high pH may be an additional controlling factor of N-fixation, as suggested by experimental nutrient additions in tropical lakes, and grazing resistance may be an additional factor of success of these large-sized phytoplankters.

**Reply:**

We also discuss those two potentially important factors in the current version of the manuscript (see lines 272-290 and 313-315).

**Anticipated changes:**

However, following this suggestion, we will now include data on zooplankton abundance in the supplements showing the lack of a relationship between grazer and diazotroph abundance.

**Comment:**

The conceptual model for Trichodesmium also involves N-fixation (with diazocytes, not heterocytes), control of growth by SRP and Fe, and vertical migration regulated by light and nutrient requirements

**Reply:**

We have tried to address some of these mechanisms in the submitted manuscript. However, given that our paper is not a review (as the publication cited by this Reviewer) we clearly do not have sufficient space or data that would allow us to discuss all aspects of the *Dolichospermum* N metabolism in the same detail as Bergman et al. (2013).

**Anticipated changes:**

We are now including the abovementioned article in the reference list.

**Comment:**

Other remarks 73-74: "*This large tropical lake is permanently stratified and characterized by a nutrient-rich hypolimnion overlaid by an oligotrophic epilimnion*"
Poor depiction of the lake ... better to mention that it is meromictic, and that the

mixolimnion is stratified during the wet season and mixed during the dry season, with a large spatial variability.

**Anticipated changes:**

We will improve this section in the revised draft and provide all relevant details as recommended.

**Comment:**

106: "*This sampling approach has been adopted in previous studies (e.g. Salonen et al., 1999; Stenuite et al., 2009)*": Yes but long time series with adequate sampling frequency are better suited to capture temporal variability, particularly when studying relatively rare "bloom" events. Combination with remote sensing data is ideal, given the very large spatial heterogeneity (see Horion et al. 2010)

**Reply:**

The timing of expeditions was pre-selected to possibly maximize our chances of encountering *Dolichospermum* blooms and we were lucky enough to observe relatively high abundances of filamentous cyanobacteria during both field campaigns. Clearly, long time series sampling coupled to remote sensing is an ideal approach to fully understand the spatiotemporal dynamics of phytoplankton blooms in any lake. Nonetheless and with full respect, this is not always possible and in the current study we aimed to resolve factors influencing the abundance of diazotrophs using the data from two lake-wide campaigns. In our efforts we attempted to sample the water column along the north-south axis (~500 km, the full expansion of the Tanzanian borders) to reflect the spatial variability. As stated in the methods, we combined a high spatial coverage (9 stations) with vertical profiles (continuous and discrete) to resolve the entire water column and measured multiple biological and environmental parameters in parallel. The two surveys have yielded two extensive data sets with synchronized measurements of phytoplankton community composition (>10 μm), pigment concentrations (Chl, PC, PE), physical parameters, and nutrient concentration data. To the best of our knowledge, a similar dataset (now also including synchronized seston stoichiometry and zooplankton counts) does not exist for Lake Tanganyika and may thus serve well for complementing previous findings and drawing some conclusions as to the studied mechanism.

**Comment:**

205: *"The phytoplankton community in Lake Tanganyika was dominated by chlorophytes, diatoms, and cyanobacteria (Fig. 2) with lower contributions from dinophytes, while euglenophytes were rare. Diazotrophic cyanobacteria, of which Dolichospermum sp. was the main taxon (we found only a few colonies of Anabaenopsis tanganyikae colonies in the South), were most abundant in the North and centre of the lake, where the primary thermocline was below the euphotic zone (Apr/May only) or deeper than ~40 m (Fig. 2 & 4)"*. Again, the statement implies that cyanobacteria were all efficient diazotrophs, which was likely not the case. Cryptophytes were overlooked.»

**Reply:**

We focused our efforts on diazotrophic cyanobacteria that are well-known and have been reported throughout decades as an important component of the phytoplankton in Lake Tanganyika. The latter part of this comment is unclear to us. Previous studies have shown that the relatively large-celled *Cryptophytes* can be rare or even absent in the phytoplankton community of Lake Tanganyika and might be subject to a long-term decline (Cocquyt and Vyverman, 2005; Verburg et al., 2003). The full dataset from our microscopic phytoplankton survey is available online and to the best of our knowledge, *Cryptophytes* do not fix $N_2$, but we welcome any more specific suggestions regarding this subject.

**Anticipated changes:**

As already stated above, we will make it very clear that the entire study is focused on medium to large sized phytoplankton fraction only.

**Comment:**

276-277 *"An additional ecological driver for diazotroph abundance may operate by mutualistic interactions between diazotrophs and diatoms"* and below (285-290). This rather wild assumption is just based on *"contact"* between cyanos and diatoms, which may have resulted from the sampling procedure. This has nothing to do with the endosymbiosis of Richelia with large diatoms in the sea.

**Reply:**

We agree with the Reviewer that this is a hypothesis based on some empirical evidence as well as supporting literature and not a firm result. We did not, however, argue to have found an endosymbiosis, but rather suggested the possibility of an

external symbiosis or a beneficial *in-situ* interaction between the two (i.e., exchange of substances in the surrounding water). The apparent physical association that we have observed was already reported by Van Meel (1954). In the context of this criticism, perhaps it may be interesting to consult the most recent literature on the topic of cyanobacteria-diatom interactions (definitely not restricted to *Richelia*), for example Foster and Zehr, 2019 or Schoffelen et al., 2019.

**Anticipated changes:**

Additional new and potentially interesting findings have also just been reported (Nieves-Morión et al., 2020) and will be referred to in the revised version.

**Comment:**

Briefly, they [Synechococcus] are not as efficient N-fixers as the heterocytous taxa, and their small size allows high growth rates and high nutrient uptake rates, which make them specialists of oligotrophic conditions. This is an issue throughout the manuscript, which does not mention these picocyanobacteria, which make 41 – 99 % of total phytoplankton biomass (Stenuite et al., JPR, 2009).

**Reply:**

While we fully agree with the need to take picocyanobacteria into consideration in a revised version of the manuscript, the focus of this study is on the well-documented main N fixing cyanobacteria (*Dolichospermum* and *Anabaenopsis*). For reasons outlined below, we argue that the presence/abundance of *Synechococcus* do not contradict or distort the mechanism observed in the current study. Thus, regardless of the picocyanobacteria distribution, we are still in full support of our conclusions.

**Comment:**

330: "*Our results also show that the fluorometric determination of extracted phycocyanin and phycoerythrin provides an excellent proxy for the abundance of filamentous cyanobacteria*" This statement is plainly false, as both phycobilins are also present in picocyanobacteria, which dominate the phytoplankton in the present lake. The fact that PC and PE concentration correlated with filamentous cyanobacteria abundance was very likely pure chance.

**Reply:**

The leading junior author, as well as all co-authors, very rarely have been able to identify such strong relationships ($p < 0.001$, Pearson correlation coefficients: 0.52-0.88) by pure chance. This is rather uncommon and would be highly unlikely in a data set comprising 2 x 9 vertical profiles during two different seasons. By contrast, in many previous studies, including the ones referred to in the previous remarks and suggestions by the Reviewer, pigments (fractionated or not) have been used in combination with flow cytometry to infer the complete structure (and biomass) of the total phytoplankton community in this lake (i.e., Stenuite et al., 2009). We are therefore confused as to the origin of the very strong statement that "*phycocyanin and phycoerythrin provides an excellent proxy for the abundance of filamentous cyanobacteria*" is false. Even through constrained to the medium to large cell fraction, this is the first study combining detailed microscopic counts of the phytoplankton community with extracted phycocyanin and phycoerythrin concentrations in Lake Tanganyika and all those data are now open access and fully available as a supplement to the current submission.

The potential of using these two pigments for identifying both filamentous and picocyanobacteria was already recognized by Salonen et al. (1999): "*Because of the important role of filamentous as well as picoplanktonic cyanobacteria in Lake Tanganyika, techniques based on their specific pigments, such as fluorescence of phycocyanin (e.g. Watras & Baker, 1988) or phycoerythrin, are likely useful for large scale studies of phytoplankton.*" Numerous studies have shown that *Dolichospermum* is rich both in phycoerythrin and phycocyanin (e.g. Moreno et al., 2003; Rodriguez et al., 1989; Watras and Baker, 1988) with recent evidence showing that C-phycocyanin and allophycocyanin together can amount to ~17 % and C-phycoerythrin to ~1 % of its biomass (Nath et al., 2020). The large colonies (we measured ~40-50 cells on average) which occurred in high densities (up to 29 colonies mL$^{-1}$) during our surveys must leave a distinct phycocyanin and phycoerythrin signal, regardless of the distribution of the small celled picocyanobacteria.

Generally, *Synechococcus* is associated with relatively high nutrient conditions in marine environments (Rajaneesh et al., 2020) as well as in Lake Tanganyika (as also insinuated by the Reviewer) and most successful in the south during the dry season (e.g. Descy et al., 2005 & 2010). Therefore, we are inclined to argue that picocyanobacteria are not dominant when nutrients (P or N) are even lower than usual and their availability strongly limited. This is the scenario we want to

describe and study in the current work. We are convinced that our study is not in contradiction to previous investigations and for above outlined reasons can constitute a complementary contribution to the existing literature on Lake Tanganyika.

**Anticipated changes:**

We will report on the importance of picocyanobacteria in Lake Tanganyika and discuss to what extent picocyanobacteria may have contributed to the measured pigment concentrations including the abovementioned references. We will argue that, based on the consistent findings from studies of the nano- and pico-fraction of the phytoplankton presented above, we would not expect picocyanobacteria to be dominant in the scenarios where filamentous, diazotrophic cyanobacteria appeared in high densities during our surveys.

**Comment:**

Table S2 contains several misspelling of taxa names, likely some wrong identifications and a mis-classification (Sphinctosiphon is a cyanoprokaryote, not a green alga).

**Anticipated changes:**

We will take great care to address the misclassification as well as any misspelled names.

**References**

Bergman, B., Sandh, G., Lin, S., Larsson, J. and Carpenter, E. J.: Trichodesmium – a widespread marine cyanobacterium with unusual nitrogen fixation properties, FEMS Microbiol Rev, 37, 286–302, doi:10.1111/j.1574-6976.2012.00352.x, 2013.

Bootsma, H. A. and Hecky, R. E.: Conservation of the African great lakes: A limnological perspective, Conserv. Biol., 7(3), 644–656, doi:10.1046/j.1523-1739.1993.07030644.x, 1993.

Cocquyt, C. and Vyverman, W.: Phytoplankton in Lake Tanganyika: a Comparison of Community Composition and Biomass off Kigoma with Previous Studies 27 Years Ago, J. Great Lakes Res., 31(4), 535–546, doi:10.1016/S0380-1330(05)70282-3, 2005.

Descy, J. P., Hardy, M. A., Sténuite, S., Pirlot, S., Leporcq, B., Kimirei, I., Sekadende, B., Mwaitega, S. R. and Sinyenza, D.: Phytoplankton pigments and community composition in Lake Tanganyika, Freshw. Biol., 50(4), 668–684, doi:10.1111/j.1365-2427.2005.01358.x,

2005.

Descy, J. P., Tarbe, A. L., Stenuite, S., Pirlot, S., Stimart, J., Vanderheyden, J., Leporcq, B., Stoyneva, M. P., Kimirei, I., Sinyinza, D. and Plisnier, P. D.: Drivers of phytoplankton diversity in Lake Tanganyika, Hydrobiologia, 653(1), 29–44, doi:10.1007/s10750-010-0343-3, 2010.

Foster, R. A. and Zehr, J. P.: Diversity, Genomics, and Distribution of Phytoplankton-Cyanobacterium Single-Cell Symbiotic Associations, Annu. Rev. Microbiol., (73), 435–56, doi:https://doi.org/10.1146/annurev-micro-090817-062650, 2019.

Grasshoff, K., Kremling, K. and Ehrhardt, M.: Methods of seawater analysis, Wiley, New York., 1999.

Hecky, R. E., Spigel, R. H., & Coulter, G. W.: The nutrient regime, in Lake Tanganyika and its life, edited by G. W. Coulter, pp. 76–89, Oxford University Press, Oxford, England., 1991.

Hecky, R. E. and Kling, H. J.: The phytoplankton and protozooplankton Lake Tanganyika: Species composition, biomass, chlorophyll content, and spatio-temporal distribution, Limnol. Ocean., 26(3), 548–564, 1981.

Hecky, R. E., Bootsma, H. A., Mugidde, R. M. and Bugenyi, F. W. B.: Phosphorous Pumps, Nitrogen Sinks, and Silcon drains Plumbing Nutrients in the African Great Lakes, in The limnology, climatology and paleoclimatology of the East African lakes, pp. 205–233, Gordon and Breach Publishers, Amsterdam, Amsterdam., 1996.

Horion, S., Bergamino, N., Stenuite, S., Descy, J. P., Plisnier, P. D., Loiselle, S. A. and Cornet, Y.: Optimized extraction of daily bio-optical time series derived from MODIS/Aqua imagery for Lake Tanganyika, Africa, Remote Sens. Environ., 114(4), 781–791, doi:10.1016/j.rse.2009.11.012, 2010.

Langenberg, V. T., Mwape, L. M., Tshibangu, K., Tumba, J.-M., Koelmans, A. A., Roijackers, R., Salonen, K., Sarvala, J., Mölsä, H., Mwape, L. M., Tshibangu, K. and Tumba, J.: Comparison of thermal stratification, light attenuation, and chlorophyll- a dynamics between the ends of Lake Tanganyika, Aquat. Ecosyst. Health Manag., 5(3), 255–265, doi:10.1080/1463498029003195, 2002.

Moreno, J., Vargas, M. Á., Rodríguez, H., Rivas, J. and Guerrero, M. G.: Outdoor cultivation of a nitrogen-fixing marine cyanobacterium, Anabaena sp. ATCC 33047, Biomol. Eng., 20(4–6), 191–197, doi:10.1016/S1389-0344(03)00051-0, 2003.

Narita, T., Mulimbwa, N. and Mizuno, T.: Vertical Distribution and Seasonal Abundance of Zooplankters in Lake Tanganyika, Afr. Study Monogr., 6, 1–16, 1986.

Nath, P. C., Tiwari, O. N., Devi, I., Bandyopadhyay, T. K. and Bhunia, B.: Biochemical and morphological fingerprints of isolated Anabaena sp.: a precious feedstock for food additives, Biomass Convers. Biorefinery, doi:10.1007/s13399-020-00651-y, 2020.

Nieves-Morión, M., Flores, E. and Foster, R. A.: Predicting substrate exchange in marine diatom-heterocystous cyanobacteria symbioses, Environ. Microbiol., 22(6), 2027–2052, 2020.

Rajaneesh, K. M., Naik, R. K., Roy, R. and D'Costa, P. M.: Cyanobacteria in tropical and subtropical marine environments: bloom formation and ecological role, INC., 2020.

Rodriguez, H., Rivas, J., Guerrero, M. G. and Losada, M.: Nitrogen-Fixing Cyanobacterium with a High Phycoerythrin Content, Appl. Environ. Microbiol., 55(3), 758–760, doi:10.1128/aem.55.3.758-760.1989, 1989.

Salonen, K., Sarvala, J., Jarvinen, M., Langenberg, V., Nuottajarvi, M., Vuorio, K. and Chitamwebwa, D. B. R.: Phytoplankton in Lake Tanganyika - vertical and horizontal distribution of in vivo fluorescence, Hydrobiologia, 407, 89–103, doi:10.1023/a:1003764825808, 1999.

Schnetger, B. and Lehners, C.: Determination of nitrate plus nitrite in small volume marine water samples using vanadium(III)chloride as a reduction agent, Mar. Chem., 160, 91–98, doi:10.1016/j.marchem.2014.01.010, 2014.

Schoffelen, N. J., Mohr, W., Ferdelman, T. G., Ploug, H. and Kuypers, M. M. M.: Phosphate availability affects fi xed nitrogen transfer from diazotrophs to their epibionts, ISME J., doi:10.1038/s41396-019-0453-5, 2019.

Stenuite, S., Pirlot, S., Hardy, M.-A., Sarmento, H., Tarbe, A.-L., Leporcq, B. and Descy, J.-P.: Phytoplankton production and growth rate in Lake Tanganyika: evidence of a decline in primary productivity in recent decades, Freshw. Biol., 52(11), 2226–2239, doi:10.1111/j.1365-2427.2007.01829.x, 2007.

Verburg, P., Hecky, R. E. and Kling, H.: Ecological consequences of a century of warming in Lake Tanganyika, Science, 301(5632), 505–507, 2003.

Vuorio, K., Nuottajärvi, M., Salonen, K. and Sarvala, J.: Spatial distribution of phytoplankton and picocyanobacteria in Lake Tanganyika in March and April 1998, Aquat. Ecosyst. Heal. Manag., 6(3), 263–278, doi:10.1080/14634980301494, 2003.

Watras, C. J. and Baker, A. L.: Detection of planktonic cyanobacteria by tandem in vivo fluorometry, Hydrobiologia, 169(1), 77–84, doi:10.1007/BF00007935, 1988.

---

## Referee Comment (RC2) · Anonymous Referee #2 · 6 Sep 2020

This study presents a very partial view on phytoplankton of Lake Tanganyika as only large phytoplankton (>10 um) was analyzed, while it has been demonstrated more than the half of phytoplankton biomass can not be counted correctly in an inverted microscope because of it small size.

The reader gets the impression that this study deals with the whole phytoplankton community, with statements like "filamentous genera Dolichospermum and Anabaenopsis, are key players under these conditions (up to 41.7 % of phytoplankton community)". This is not true because picophytoplankton, which accounts for >50% (up to 80%) of phytoplankton biomass, was totally ignored in this study. For example Fig. 2 gives the

impression that phytoplankton is dominated by chlorophytes, which is not true. This is actually reinforced in the text line 207:"The phytoplankton community in Lake Tanganyika was dominated by chlorophytes, diatoms, and cyanobacteria (Fig. 2)". It is not true!

The authors focus their discussion on N limitation, but P is also a main limiting factor in such oligotrophic systems. Why N and not P? The literature on East African Great lakes suggests that P is actually the main limiting factor!

Another major fragility of this work is that the authors draw conclusions based only on circumstantial observations, linking nutrient concentration profiles with microscope observations of phytoplankton >10um. Their conclusions are not supported by any experiment nor statistical analysis. Taking into account that ecological processes in Lake Tanganyika are totally dominated by microbial compartments smaller than 10um, which were not took into account in this study, their conclusions probably do not stand.

---

## Author Comment (AC2) · 27 Sep 2020

We thank the reviewer for commenting on our manuscript and for all the constructive feedback that helps to improve our manuscript. We have noticed that this review reiterates the points made by reviewer #1. Hence, we have concretized some of the changes in the manuscript here.

We have addressed the comments starting with "Reply:".

**Comment:**

This study presents a very partial view on phytoplankton of Lake Tanganyika as only large phytoplankton (>10 um) was analyzed, while it has been demonstrated more than the half of phytoplankton biomass can not be counted correctly in an inverted microscope because of it small size. The reader gets the impression that this study deals with the whole phytoplankton community, with statements like "filamentous genera Dolichospermum and Anabaenopsis, are key players under these conditions (up to 41.7 % of phytoplankton community)". This is not true because picophytoplankton, which accounts for >50% (up to 80%) of phytoplankton biomass, was totally ignored in this study. For example Fig. 2 gives the impression that phytoplankton is dominated by chlorophytes, which is not true. This is actually reinforced in the text line 207:"The phytoplankton community in Lake Tanganyika was dominated by chlorophytes, diatoms, and cyanobacteria (Fig. 2)". It is not true!.

**Reply:**

We acknowledge that the wording concerning the studied part of the phytoplankton community must be more precise. However, the goal of this study was to investigate

N fixing, filamentous cyanobacteria and not to analyze the diversity of the entire phytoplankton community as well as its controls. Blooms of filamentous cyanobacteria were frequently observed in Lake Tanganyika, but have been – to the best of our knowledge – never comprehensively studied. Hence, the data and analyses in the current work provide novel insight into factors potentially controlling their abundance in Lake Tanganyika, which may stimulate further research to towards understanding the future of diazotrophic cyanobacteria in an increasingly stratified lake (Verburg et al., 2003; Verburg and Hecky, 2009) as well as other deep, oligotrophic (sub)tropical water bodies.

A conservative biovolume estimate (assuming the same biovolume for cells of the smaller picocyanobacteria and filamentous cyanobacteria) based on metagenomic analyses from stations 2 and 7 (Apr/May) suggests that filamentous taxa make up ~50% of the cyanobacterial biovolume in our samples. We will present the metagenomic analyses in a forthcoming article, but data can be made available upon request for Reviewers/Editor.

**Anticipated changes:**
In the revised version, we make it very clear that the entire study is focused on the larger, microscopic (>10 μm) phytoplankton fraction and that we cannot make any strong assertions about either smaller fractions or about the total phytoplankton biomass in the lake. Line 207 reads now "The large fraction (>10 μm) of the phytoplankton community in Lake Tanganyika, was dominated …". At all other parts, where phytoplankton community was mentioned, we added "(>10 μm)" or "the studied/large fraction of the phytoplankton community" now. The axis labels of Fig. 2 were changed to "Abundance of >10 μm fraction …" and "Relative abundance of >10 μm fraction …". We have also updated Fig. 4 (see below).

Further, we have added a discussion of picoplankton and phycobilin pigments:
"The phycoerythrin (PE) and phycocyanin (PC) concentrations will also be influenced by picocyanobacteria, which are known to contain at least PE in Lake Tanganyika (Stenuite et al., 2009). It is important to underline here that picoplankton usually dominates the total phytoplankton abundance and biovolume in Lake Tanganyika, especially in the south of the lake and during the dry season (Descy et al., 2005, 2010;

Stenuite et al., 2009; De Wever et al., 2005). Picocyanobacteria are associated with relatively high nutrient conditions in Lake Tanganyika (e.g. Descy et al., 2005 & 2010; Stenuite et al., 2009) and decrease in abundance when filamentous cyanobacteria thrive (see Fig. 5 in Descy et al., 2010). During our surveys, changes in PC and PE concentrations were strongly correlated with the abundances of filamentous cyanobacteria ($p < 0.001$, Pearson correlation coefficients: 0.52-0.88). Thus, we have interpreted the near-surface peaks in PC and PE at stations 4-6 (Sep/Oct) as well as 2-6 (Apr/May) as coming from filamentous, diazotrophic cyanobacteria with possibly lower contributions from picocyanobacteria. The simple cell abundance-PC relationship from Kong et al. (2014) shows that *Dolichospermum* can indeed be responsible for a large fraction of PC measured in this study. Using the slope of the regression line, we estimated a PC concentration of 0.90 µg L$^{-1}$ for the PC and *Dolichospermum* maximum (upper 25 m at station 3), representing 65 % of the measured 1.38 µg L$^{-1}$. By contrast, at stations without filamentous cyanobacteria, the phycobilin pigment signals must originate from *Synechococcus* and other small cyanobacteria such as *Microcystis* and *Chrococcus*. Phycocyanin concentrations were often below detection limit in the south and at least an order of magnitude lower in the north (max. 0.05 versus max. 1.67 µg L$^{-1}$) compared to the surface peaks associated to the presence of filamentous cyanobacteria (Fig. 1). Aside from stations 4-6 (Sep/Oct) as well as 2-6 (Apr/May), PE typically formed subsurface maxima corresponding to the chlorophyll-a peak (Fig. 1 & XXX). Noteworthy, PE occurred at concentrations of at least 0.003 µg L$^{-1}$ at all stations including the south basin, where picocyanobacteria are known to dominate (Descy et al., 2005 & 2010; Stenuite et al., 2009). Thus, we argue that PE rather than PC represents contributions from picocyanobacteria."

Following this discussion, we have changed L. 331 to: "Our results also show that the fluorometric determination of extracted phycoerythrin and especially phycocyanin can provide a suitable proxy for filamentous cyanobacteria, when they occur in high densities."

[Figure]

Fig. XXX: Phycoerythrin concentrations during (a) the end of the dry season 2017 (Sep/Oct) and (b) the end of the rainy season 2018 (Apr/May) in Lake Tanganyika.

**Comment:**

The authors focus their discussion on N limitation, but P is also a main limiting factor in such oligotrophic systems. Why N and not P? The literature on East African Great lakes suggests that P is actually the main limiting factor!

**Reply:**

As indicated by the reviewer, the nutrient limitation in Lake Tanganyika may be complex and spatiotemporalily variable. The seston stoichiometry and bioassay studies in this lake suggest that the nutrient supply is relatively balanced with P limitation occurring more often than N limitation (Edmond et al., 1993; Järvinen et al., 1999; North et al., 2008; Stenuite et al., 2007; De Wever et al., 2008). By contrast, the dissolved nutrient N:P ratios are typically well below Redfield ratio (Descy et al., 2010; Edmond et al., 1993). Early limnological studies have already recognized this discrepancy and argued that inputs from N fixation may potentially sustain the balanced nutrient supply (Bootsma and Hecky, 1993; Hecky, R. E., Spigel, R. H., & Coulter, 1991). Despite the potential importance, N fixation and factors controlling key diazotrophs (i.e., filamentous cyanobacteria) have not been directly studied in Lake Tanganyika.

The prevalence of N limitation during our study was supported not only by the relative N deficit, but also the fact that at 8 out of 9 stations (Apr/May) and 6 out of 9 stations (Sep/Oct), the levels of free nitrate in the surface waters were below the detection limit and more often so when the thermocline was below the euphotic zone (i.e., the supply of nitrate from deeper waters was low). In addition, we have now analyzed the seston N:P ratios, pointing towards frequent N limitation during our surveys (see below). Last but not least, phytoplankton taxa have different requirements and may therefore be limited at different nutrient levels and ratios (De Wever et al., 2008). Our study organisms (filamentous cyanobacteria) are distinguished by their capability of fixing N. Given the fact that N fixation is known to be metabolically expensive and that we found evidence for N fixation (see below) in the surface waters, we argue that phytoplankton was indeed N limited during our study.

**Anticipated changes:**

We are presenting now data on the seston N:P ratios that, according to Stenuite et al. (2007), are indicative of frequent phytoplankton N limitation during our survey (see below) and support our reasoning.

The respective part in the results, line 177-182 reads now as follows:
"Nitrate was the main form of DIN accessible to phytoplankton (Fig. 1), while $NH_4^+$ and $NO_2^-$ remained below detection limit in the upper 120-150 m. The euphotic zone, characterized in Apr/May, varied between 35.8-54.8 m with an average of 47.3 m. Nitrate concentrations in the euphotic zone (Fig. 1 b,e) were also often below detection limit while $PO_4^{3-}$ concentrations were relatively high (Fig. 3 & S2). The DIN-depleted euphotic zone, the N deficit (98 % of all observations) persisting throughout the water column (Fig. S2), and seston N:P ratios typically below 22 (Fig. YYY) together imply that primary productivity was likely N limited during our surveys (Guildford and Hecky, 2000; Healey and Hendzel, 1979; Stenuite et al., 2007)."

[Figure]

Fig. YYY: The molar N:P ratios of seston during the end of the dry season 2017 (Sep/Oct) and the end of the rainy season 2018 (Apr/May) in Lake Tanganyika. Data is shown for the upper 50 m, encompassing the euphotic zone and the deep chlorophyll maximum. The dotted red line represents the cut-off for N or P limitation defined by previous studies (Guildford and Hecky, 2000; Healey and Hendzel, 1979; Stenuite et al., 2007), with N limitation occurring at ratios <22 and P limitation at ratios >22.

**Comment:**

Another major fragility of this work is that the authors draw conclusions based only on circumstantial observations, linking nutrient concentration profiles with microscope observations of phytoplankton >10um. Their conclusions are not supported by any experiment nor statistical analysis. Taking into account that ecological processes in Lake Tanganyika are totally dominated by microbial compartments smaller than 10um, which were not took into account in this study, their conclusions probably do not stand.

**Reply:**

We fully agree with the need acknowledge the importance of picocyanobacteria in a revised version of this manuscript. The focus of this study was, however, on N fixing,

filamentous cyanobacteria (*Dolichospermum* and *Anabaenopsis*) and factors potentially regulating their abundances. The data and analyses presented in our work address this goal and support our conclusions (N fixing cyanobacteria reach higher numbers as a result of reduced $NO_3^-$ fluxes, when free P is available).

Blooms of filamentous cyanobacteria were reported from multiple phytoplankton surveys in Lake Tanganyika (Cocquyt and Vyverman, 2005; Descy et al., 2005, 2010; Hecky and Kling, 1981; Langenberg et al., 2002; Narita et al., 1986; Salonen et al., 1999; Vuorio et al., 2003). While filamentous cyanobacteria may not be the dominant component of the phytoplankton community in terms of biomass during a large part of the year, they likely are of disproportionate ecological importance when occurring in high numbers. They are the only known N fixer and thus, add freshly fixed N to the oligotrophic and highly N deficient waters of Lake Tanganyika.

The metagenomic analyses conducted during Apr/May reveal that *Synechococcus* in Lake Tanganyika do not contain any regulatory genes of the nitrogenase complex and thus, do not have the capability to fix N. On the other hand, the presence of abundant heterocysts in *Dolichospermum* colonies supports active N fixation (Fig. 5). These observations are further substantiated by $^{15\text{-}15}N_2$ incubation experiments, which yielded high maximum N fixation rates of $\geq 5$ nmol N $L^{-1}$ $d^{-1}$ at stations where filamentous cyanobacteria were abundant. Filamentous cyanobacteria and N fixation rates were significantly correlated ($p < 0.05$, $R^2 = 0.80$). We will present the metagenomic and incubation data in a forthcoming article (data can be made available upon request for Reviewers/Editor).

We are convinced that the analysis presented in Fig. 4 is more transparent than a simplified statistical model, which does not respect the spatial autocorrelation and i.e. dependence of samples. Instead, Fig. 4 visualizes a clear pattern and the variability between individual stations. The analysis in Fig. 4 a is additionally supported by a breakpoint model estimating a mean threshold thermocline depth (36.7 m) that matches well with the average euphotic zone thickness of around 40 m (Cocquyt and Vyverman, 2005; Descy et al., 2005; Hecky et al., 1978), further supporting the results in Fig. 4 b.

**Anticipated changes:**

We have now removed any implications that the studied size fraction of the phytoplankton community may control the overall phytoplankton abundance or biomass (see above). To further enhancing clarity, we present the absolute instead of relative abundances of filamentous cyanobacteria in an updated version of Fig. 4 (see below). Using the total abundances no longer required treating station 9 (Sep/Oct) as an outlier in the breakpoint model due to the larger absolute changes, making the results even more convincing in our opinion. The breakpoint in panel (a) shifted only marginally, yielding the same rounded value of 36.7 m.

[Figure]

**References**

Bootsma, H. A. and Hecky, R. E.: Conservation of the African great lakes: A limnological perspective, Conserv. Biol., 7(3), 644–656, doi:10.1046/j.1523-1739.1993.07030644.x, 1993.

Cocquyt, C. and Vyverman, W.: Phytoplankton in Lake Tanganyika: a Comparison of Community Composition and Biomass off Kigoma with Previous Studies 27 Years Ago, J. Great Lakes Res., 31(4), 535–546, doi:10.1016/S0380-1330(05)70282-3, 2005.

Descy, J. P., Hardy, M. A., Sténuite, S., Pirlot, S., Leporcq, B., Kimirei, I., Sekadende, B., Mwaitega, S. R. and Sinyenza, D.: Phytoplankton pigments and community composition in Lake Tanganyika, Freshw. Biol., 50(4), 668–684, doi:10.1111/j.1365-2427.2005.01358.x, 2005.

Descy, J. P., Tarbe, A. L., Stenuite, S., Pirlot, S., Stimart, J., Vanderheyden, J., Leporcq, B., Stoyneva, M. P., Kimirei, I., Sinyinza, D. and Plisnier, P. D.: Drivers of phytoplankton diversity in Lake Tanganyika, Hydrobiologia, 653(1), 29–44, doi:10.1007/s10750-010-0343-3, 2010.

Edmond, J. M., Stallard, R. F., Craig, H., Craig, V., Weiss, R. F. and Coulter, G. W.: Nutrient chemistry of the water column of Lake Tanganyika, Limnol. Oceanogr., 38(4), 725–738, doi:10.4319/lo.1993.38.4.0725, 1993.

Guildford, S. J. and Hecky, R. E.: Total nitrogen, total phosphorus, and nutrient limitation in lakes and oceans: Is there a common relationship?, Limnol. Oceanogr., 45(6), 1213–1223, doi:10.4319/lo.2000.45.6.1213, 2000.

Healey, F. P. and Hendzel, L. L.: Indicators of Phosphorus and Nitrogen Deficiency in Five Algae in Culture, J. Fish. Res. Board Canada, 36(11), 1364–1369, doi:10.1139/f79-195, 1979.

Hecky, R. E., Spigel, R. H., & Coulter, G. W.: The nutrient regime, in Lake Tanganyika and its life, edited by G. W. Coulter, pp. 76–89, Oxford University Press, Oxford, England., 1991.

Hecky, R. E. and Kling, H. J.: The phytoplankton and protozooplankton Lake Tanganyika: Species composition, biomass, chlorophyll content, and spatio-temporal distribution, Limnol. Ocean., 26(3), 548–564, 1981.

Hecky, R. E., Fee, E. J., Kling, H. and Rudd, J. W. M.: Studies on the planktonic ecology of Lake Tanganyika, Winnipeg, Manitoba., 1978.

Järvinen, M., Salonen, K., Sarvala, J., Vuorio, K. and Virtanen, A.: The stoichiometry of particulate nutrients in Lake Tanganyika - Implications for nutrient limitation of phytoplankton, Hydrobiologia, 407, 81–88, doi:10.1023/A:1003706002126, 1999.

Kong, Y., Lou, I., Zhang, Y., Lou, C. U. and Mok, K. M.: Using an online phycocyanin fluorescence probe for rapid monitoring of cyanobacteria in Macau freshwater reservoir, Hydrobiologia, 741(1), 33–49, doi:10.1007/s10750-013-1759-3, 2014.

Langenberg, V. T., Mwape, L. M., Tshibangu, K., Tumba, J.-M., Koelmans, A. A., Roijackers, R., Salonen, K., Sarvala, J., Mölsä, H., Mwape, L. M., Tshibangu, K. and Tumba, J.: Comparison of thermal stratification, light attenuation, and chlorophyll- a dynamics between the ends of Lake Tanganyika, Aquat. Ecosyst. Health Manag., 5(3), 255–265, doi:10.1080/1463498029003195, 2002.

Narita, T., Mulimbwa, N. and Mizuno, T.: Vertical Distribution and Seasonal Abundance of Zooplankters in Lake Tanganyika, Afr. Study Monogr., 6, 1–16, 1986.

North, R. L., Guildford, S. J., Smith, R. E. H., Twiss, M. R. and Kling, H. J.: Nitrogen,

phosphorus, and iron colimitation of phytoplankton communities in the nearshore and offshore regions of the African Great Lakes, Int. Vereinigung für Theor. und Angew. Limnol. Verhandlungen, 30(2), 259–264, doi:10.1080/03680770.2008.11902122, 2008.

Salonen, K., Sarvala, J., Jarvinen, M., Langenberg, V., Nuottajarvi, M., Vuorio, K. and Chitamwebwa, D. B. R.: Phytoplankton in Lake Tanganyika - vertical and horizontal distribution of in vivo fluorescence, Hydrobiologia, 407, 89–103, doi:10.1023/a:1003764825808, 1999.

Stenuite, S., Pirlot, S., Hardy, M.-A., Sarmento, H., Tarbe, A.-L., Leporcq, B. and Descy, J.-P.: Phytoplankton production and growth rate in Lake Tanganyika: evidence of a decline in primary productivity in recent decades, Freshw. Biol., 52(11), 2226–2239, doi:10.1111/j.1365-2427.2007.01829.x, 2007.

Stenuite, S., Tarbe, A. L., Sarmento, H., Unrein, F., Pirlot, S., Sinyinza, D., Thill, S., Lecomte, M., Leporcq, B., Gasol, J. M. and Descy, J. P.: Photosynthetic picoplankton in Lake Tanganyika: Biomass distribution patterns with depth, season and basin, J. Plankton Res., 31(12), 1531–1544, doi:10.1093/plankt/fbp090, 2009.

Verburg, P. and Hecky, R. E.: The physics of the warming of Lake Tanganyika by climate change, Limnol. Oceanogr., 54(6 PART 2), 2418–2430, doi:10.4319/lo.2009.54.6_part_2.2418, 2009.

Verburg, P., Hecky, R. E. and Kling, H.: Ecological consequences of a century of warming in Lake Tanganyika, Science, 301(5632), 505–507, 2003.

Vuorio, K., Nuottajärvi, M., Salonen, K. and Sarvala, J.: Spatial distribution of phytoplankton and picocyanobacteria in Lake Tanganyika in March and April 1998, Aquat. Ecosyst. Heal. Manag., 6(3), 263–278, doi:10.1080/14634980301494, 2003.

De Wever, A., Muylaert, K., Van Der Gucht, K., Pirlot, S., Cocquyt, C., Descy, J. and Plisnier, P.: Bacterial Community Composition in Lake Tanganyika : Vertical and Horizontal Heterogeneity, Society, 71(9), 5029–5037, doi:10.1128/AEM.71.9.5029, 2005.

De Wever, A., Muylaert, K., Langlet, D., Alleman, L., Descy, J. P., André, L., Cocquyt, C. and Vyverman, W.: Differential response of phytoplankton to additions of nitrogen, phosphorus and iron in Lake Tanganyika, Freshw. Biol., 53(2), 264–277, doi:10.1111/j.1365-2427.2007.01890.x, 2008.